# Specialized neurons in the right habenula mediate response to aversive olfactory cues

**Jung-Hwa Choi[1†], Erik R Duboue[2,3], Michelle Macurak[1], Jean-Michel Chanchu[1], Marnie E Halpern[1*†]**

[1]Carnegie Institution for Science, Department of Embryology, Baltimore, United States; [2]Jupiter Life Science Initiative, Florida Atlantic University, Jupiter, United States; [3]Wilkes Honors College, Florida Atlantic University, Jupiter, United States

**\*For correspondence:**
Marnie.E.Halpern@Dartmouth.edu

**Present address:** [†]Geisel School of Medicine at Dartmouth, Department of Molecular and Systems Biology, Hanover, United States

**Competing interest:** The authors declare that no competing interests exist.

**Abstract** Hemispheric specializations are well studied at the functional level but less is known about the underlying neural mechanisms. We identified a small cluster of cholinergic neurons in the dorsal habenula (dHb) of zebrafish, defined by their expression of the *lecithin retinol acyltransferase domain containing 2* a (*lratd2a*) gene and their efferent connections with a subregion of the ventral interpeduncular nucleus (vIPN). The *lratd2a*-expressing neurons in the right dHb are innervated by a subset of mitral cells from both the left and right olfactory bulb and are activated upon exposure to the odorant cadaverine that is repellent to adult zebrafish. Using an intersectional strategy to drive expression of the botulinum neurotoxin specifically in these neurons, we find that adults no longer show aversion to cadaverine. Mutants with left-isomerized dHb that lack these neurons are also less repelled by cadaverine and their behavioral response to alarm substance, a potent aversive cue, is diminished. However, mutants in which both dHb have right identity appear more reactive to alarm substance. The results implicate an asymmetric dHb-vIPN neural circuit in the processing of repulsive olfactory cues and in modulating the resultant behavioral response.

## Editor's evaluation

This work presents a conceptual advance on our understanding of the habenula in vertebrate species, by revealing interesting functions to specific cell types within this region of the brain.

## Introduction

Fish use the sense of smell to search for food, detect danger, navigate, and communicate social information by detecting chemical cues in their aquatic environment (*Yoshihara, 2014*). As with birds and mammals, perception of olfactory cues is lateralized and influences behavior (*Siniscalchi, 2017*). In zebrafish, nine glomerular clusters in the olfactory bulb (OB) receive olfactory information from sensory neurons in the olfactory epithelium and, in turn, transmit signals to four forebrain regions: the posterior zone of the dorsal telencephalon (Dp), the ventral nucleus of the ventral telencephalon (Vv), the posterior tuberculum (PT), and the dorsal habenular region (dHb) (*Miyasaka et al., 2014*; *Yoshihara, 2014*). In contrast to all other target regions, which are located on both sides of the forebrain, only the right nucleus of the dHb is innervated by mitral cells that emanate from medio-dorsal (mdG) and ventro-medial (vmG) glomerular clusters in both OBs (*Miyasaka et al., 2014*; *Yoshihara, 2014*). Moreover, calcium imaging experiments suggest that the right dHb shows a preferential response to odorants compared to the left dHb (*Chen et al., 2019*; *Dreosti et al., 2014*; *Jetti et al., 2014*;

*Krishnan et al., 2014*). The identity of the post-synaptic neurons within the right dHb that receive olfactory input and the purpose of this asymmetric connection are unknown.

The habenulae are highly conserved structures in the vertebrate brain and, in teleosts such as zebrafish, consist of dorsal and ventral (vHb) nuclei, which are equivalent to the medial and lateral habenulae of mammals, respectively (*Amo et al., 2010*). The neurons of the dHb are largely glutamatergic and contain specialized subpopulations that also produce acetylcholine, substance P, or somatostatin (*deCarvalho et al., 2014*; *Hsu et al., 2016*; *Lee et al., 2019*). In zebrafish, the number of neurons within each subtype differs between the left and right dHb (*deCarvalho et al., 2014*). The dHb have been implicated in diverse states such as reward, fear, anxiety, sleep, and addiction (*Duboué et al., 2017*; *Hikosaka, 2010*; *Lee et al., 2019*; *Okamoto et al., 2012*). Accordingly, the right dHb was shown to respond to bile acid and involved in food-seeking behaviors (*Chen et al., 2019*; *Krishnan et al., 2014*), whereas the left dHb was found to be activated by light and attenuate fear responses (*Dreosti et al., 2014*; *Duboué et al., 2017*; *Zhang et al., 2017*). However, the properties of the dHb neurons implicated in these behaviors, such as their neurotransmitter identity and precise connectivity with their unpaired target, the midbrain interpeduncular nucleus (IPN), have yet to be determined.

Here, we describe a group of cholinergic neurons defined by their expression of the *lecithin retinol acyltransferase domain containing 2 a* (*lratd2a*) gene [formerly known as *family with sequence similarity 84 member B* (*fam84b*)], that are predominantly located in the right dHb where they are selectively innervated by the olfactory mitral cells that originate from both sides of the brain (*Miyasaka et al., 2009*), and form efferent connections with a restricted subregion of the ventral IPN (vIPN). Activity of the *lratd2a*-expressing neurons is increased following exposure to the aversive odorant cadaverine and, when these neurons are inactivated, adult zebrafish show a diminished repulsive response. Our findings provide further evidence for functional specialization of the left and right habenular nuclei and reveal the neuronal pathway that mediates a lateralized olfactory response.

## Results

### *lratd2a*-expressing neurons in the right dHb receive bilateral olfactory input

A subset of medio-dorsal mitral cells that are labeled by *Tg(lhx2a:gap-YFP)* were previously shown to project their axons bilaterally through the telencephalon and terminate in the right dHb (*Miyasaka et al., 2009*), in the vicinity of a small population of *lratd2a*-expressing neurons [(*deCarvalho et al., 2013*) and *Figure 1A*]. To characterize this subset of dHb neurons, we used CRISPR/Cas9-mediated targeted integration (*Kimura et al., 2014*) to introduce the QF2 transcription factor (*Ghosh and Halpern, 2016*; *Subedi et al., 2014*) under the control of *lratd2a* transcriptional regulation (*Figure 1B*). QF2 does not disrupt transcription at the *lratd2a* locus (*Figure 1—figure supplement 1*) and drives expression of QUAS-regulated genes encoding fluorescent protein reporters in a similar pattern to endogenous *lratd2a* gene expression in the nervous system (*deCarvalho et al., 2013*). In 5 days post fertilization (dpf) larval zebrafish, this includes a subset of neurons in the OB, the bilateral vHb, and asymmetrically distributed neurons in the dHb. There are 2.5 times more cells in the right dHb than the left (*Figure 1C–E"* and *Figure 1—figure supplement 2*) and those on the left consistently show weaker *lratd2a* expression and less GFP labeling than those on the right (*Figure 1A* and *Figure 1—figure supplements 1–2*). Double labeling confirmed that axons of the bilateral *lhx2a* positive olfactory mitral cells terminate precisely at the *lratd2a*-expressing neurons in the right dHb (*Figure 1C–D"*, *Figure 1—figure supplement 3B*), co-localized with the pre-synaptic protein synaptophysin (*Figure 1—figure supplement 3A*). The *lratd2a*-expressing dHb neurons, in turn, project to a restricted region of the ventral IPN (*Figure 1F–G"*).

### Aversive olfactory cues activate *lratd2a* neurons in the right dHb

Several studies using transgenic expression of the genetically encoded calcium indicator GCaMP have demonstrated that dHb neurons are activated by olfactory cues in larval zebrafish (*Jetti et al., 2014*; *Krishnan et al., 2014*). We had previously examined the habenular response of adult zebrafish to cadaverine, a known aversive olfactory cue that is released from decaying fish (*Hussain et al., 2013*), and to chondroitin sulfate, a component of alarm substance (also known as Schreckstoff), which is

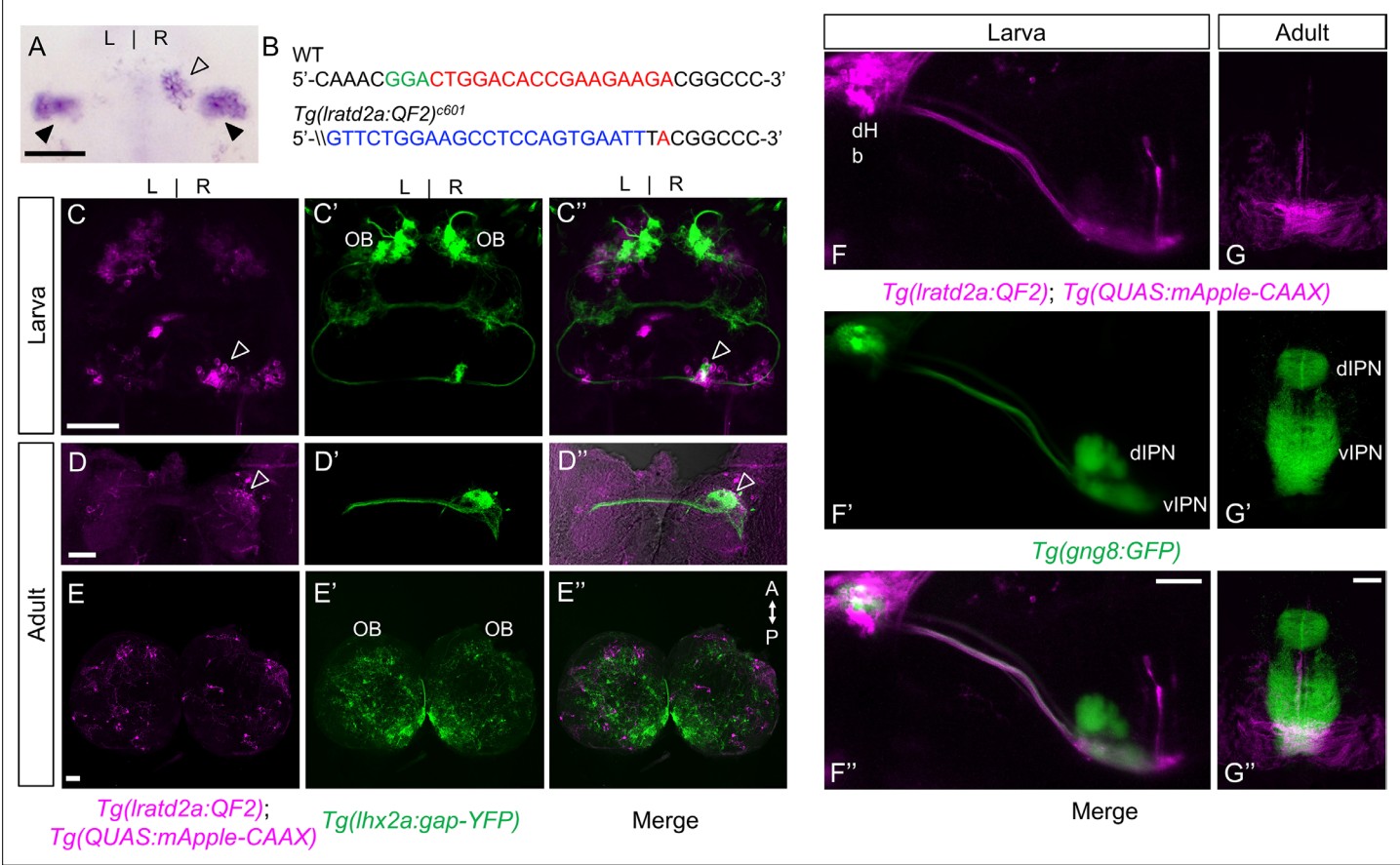

**Figure 1.** *lratd2a*-expressing neurons in the right dHb connect asymmetric pathway from the olfactory bulb to ventral IPN. (**A**) Pattern of *lratd2a* expression at 5 days post fertilization (dpf), open arrowhead indicates right dHb and black arrowheads the bilateral vHb. (**B**) Sequences of WT (top) and transgenic fish (bottom) with QF2 integrated within the first exon of the *lratd2a* gene. PAM sequences are green, the sgRNA-binding site red and donor DNA blue. Confocal dorsal views of *Tg(lratd2a:QF2), Tg(QUAS:mApple-CAAX)* and *Tg(lhx2a:gap-YFP)* labeling in a (**C-C″**) 5 dpf larva and in transverse sections of the adult brain at 3 months post-fertilization (mpf) at the level of the (**D-D″**) dHb and (**E-E″**) olfactory bulb. Axons of *lhx2a* olfactory mitral cells (open arrowheads, **C and D**) terminate at *lratd2a* dHb neurons. (**F-F″**) Lateral view of *Tg(lratd2a:QF2), Tg(QUAS:mApple-CAAX), Tg(gng8:GFP)* larva at 6 dpf with mApple-labeled dHb terminals at the ventral interpeduncular nucleus (vIPN). Dorsal habenular nuclei (dHb), dorsal interpeduncular nucleus (dIPN). (**G-G″**) Axonal endings of *lratd2a* dHb neurons are restricted to the ventralmost region of the vIPN in transverse section of 2.5 mpf adult brain. Scale bar, 50 µm. A-P, anterior to posterior; L-R, left-right; OB, olfactory bulb.

The online version of this article includes the following source data and figure supplement(s) for figure 1:

**Figure supplement 1.** Targeted genomic integration does not disrupt endogenous gene expression.

**Figure supplement 2.** Asymmetry of *lratd2a*-expressing dHb neurons.

**Figure supplement 2—source data 1.** Source data associated with *Figure 1—figure supplement 2B*.

**Figure supplement 3.** *lhx2a*-expressing olfactory cells innervate *lratd2a* dHb neurons.

released from the skin of injured fish (*Mathuru et al., 2012*). However, we did not have the necessary transgenic tools to specifically record the activity of the *lratd2a* cell population (*deCarvalho et al., 2014*).

To determine whether olfactants activate the *lratd2a*-positive neuronal cluster in the dHb, we measured calcium signaling in *Tg(lratd2a:QF2)^c644, Tg(QUAS:GCaMP6f)^c587* larvae at 7 dpf. We monitored the responses of individual cells in the right dHb, as *GCaMP6f* labeling was weakly or not detected in neurons on the left (data not shown). The observed changes in fluorescence intensity after the addition of two applications of an aversive cue were compared to those observed after two additions of vehicle alone (i.e. deionized water), all delivered 1 min apart (*Figure 2A and B*). Cadaverine elicited a significant increase in *GCaMP6f* labeling in the same neurons following the first and second applications (16-fold increase averaged over both doses; *Figure 2C*). On average, a greater than

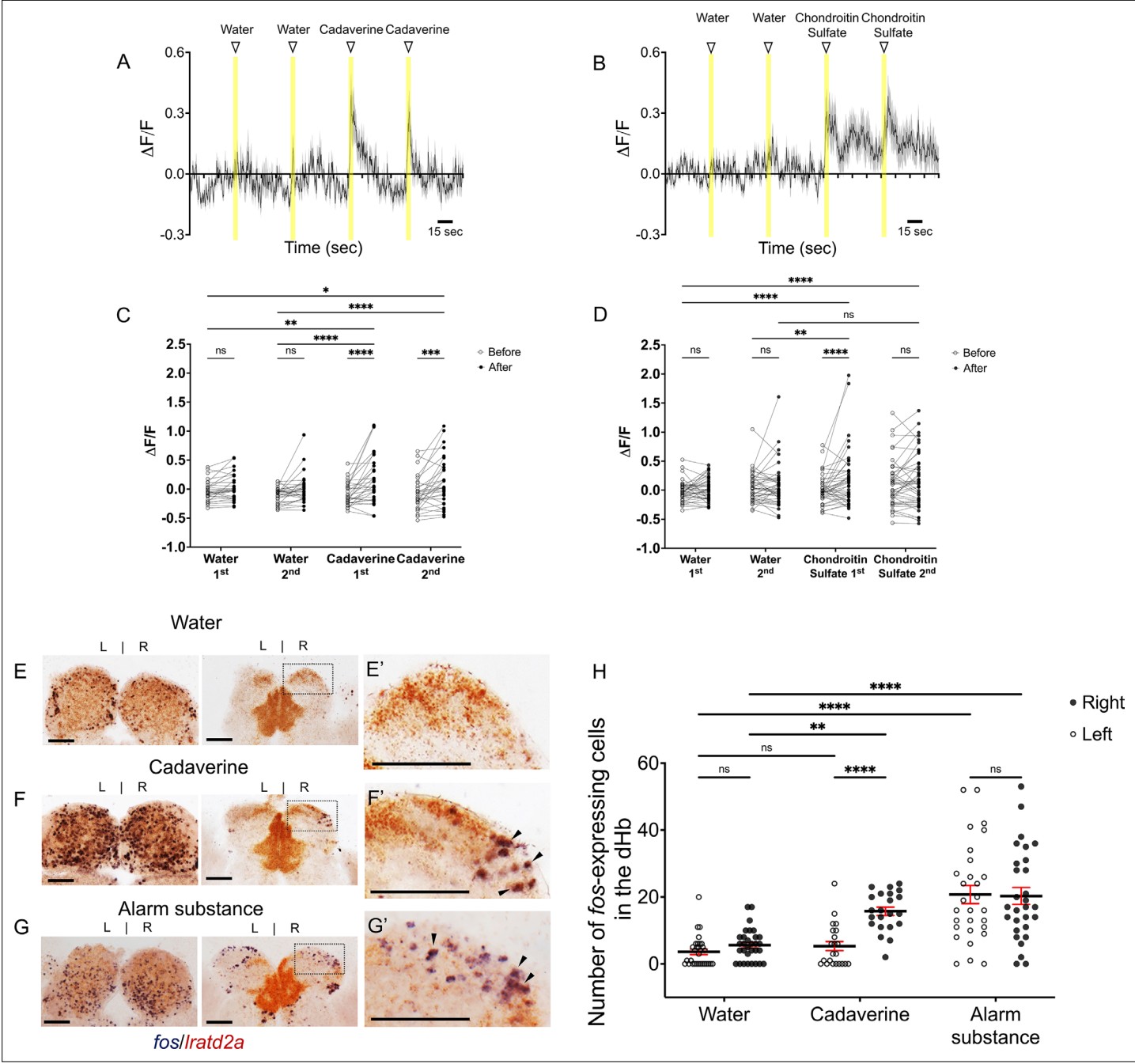

**Figure 2.** Increased activity of *lratd2a*-expressing dHb neurons upon exposure to aversive olfactory cues. (**A–B**) Average change in fluorescence (ΔF/F) in seconds (sec) for all *lratd2a* positive neurons in the larval right dHb over a 5 min interval. Yellow bars indicate consecutive 5 s intervals of vehicle or odor delivery. Solid lines represent mean responses to (**A**) cadaverine or (**B**) to chondroitin sulfate and shadings represent the standard error of the mean (SEM). (**C–D**) Change in the intensity of GCaMP6f fluorescence for *lratd2a* neurons in the right dHb in response to consecutive delivery of (**C**) water or cadaverine [n = 30 neurons in three larvae, 0.009 ± 0.039 ΔF/F for water, 0.148 ± 0.074 ΔF/F for cadaverine] and of (**D**) water or chondroitin sulfate [n = 43 neurons in four larvae, 0.039 ± 0.037 ΔF/F for water, 0.176 ± 0.069 ΔF/F for chondroitin sulfate] at seven dpf, respectively. Two-way ANOVA reveals a significant effect of time [$F_{(1, 29)}$ = 32.09, p < 0.0001], interaction [$F_{(3, 87)}$ = 3.797, p = 0.0131] but no effect of vehicle vs. odorants [$F_{(3, 87)}$ = 1.169, p = 0.3262] for cadaverine; and a significant effect of time [$F_{(1, 42)}$ = 4.754, p = 0.0349], vehicle vs. odorants [$F_{(3, 126)}$ = 2.825, p = 0.0414] and interaction [$F_{(3, 126)}$ = 4.256, p = 0.0067] for chondroitin sulfate. Post-hoc analysis by Bonferroni's multiple comparisons. (**E–G**) Colocalization of *fos* and *lratd2a* transcripts in transverse sections of adult olfactory bulbs (left panels) and habenulae (middle panels) detected by double labeling RNA in situ hybridization 30 min after addition of (**E**) water, (**F**) cadaverine, or (**G**) alarm substance to the test tank. (**E'-G'**) Higher magnification images (corresponding to dashed boxes in E-G) show *lratd2a* (brown) coexpressed with *fos* (blue) in cells of the right dHb (arrowheads). Scale bars, 100 μm. (**H**) Quantification of *fos*-expressing cells in the adult dHb after addition of water [3.58 ± 0.811 cells in the left and 5.61 ± 0.85 in the right dHb, n = 31 sections from 16 adult brains], cadaverine [5.32 ±

*Figure 2 continued on next page*

Figure 2 continued

1.36 cells in the left and 15.73 ± 1.25 in the right dHb, n = 22 sections from 11 adult brains], or alarm substance [20.72 ± 2.70 cells in the left and 20.31 ± 2.53 in the right dHb, n = 29 sections from 17 adult brains]. For the right dHb, significantly more cells were *fos* positive after addition of cadaverine (p = 0.0031) or alarm substance (p < 0.0001). For the left dHb, a significant difference was only observed after addition of alarm substance (p < 0.0001). Two-way mixed ANOVA reveals a significant effect of group [$F_{(2, 60)}$ = 30.18, p < 0.0001], left vs. right [$F_{(1, 30)}$ = 13.02, p = 0.0011] and interaction [$F_{(2, 38)}$ = 7.881, p = 0.0014]. Post-hoc analysis by Bonferroni's multiple comparisons. All numbers represent the mean ± SEM.

The online version of this article includes the following source data for figure 2:

**Source data 1.** Source data associated with *Figure 2A–D and H*.

fourfold increase in calcium signaling was measured in individual *lratd2a*-expressing neurons following addition of chondroitin sulfate compared to their response to vehicle alone (*Figure 2D*).

To examine whether the response to aversive odorants persists in the olfactory-dHb pathway of adult zebrafish, we used expression of the *fos* gene as an indicator of neuronal activation (*deCarvalho et al., 2013*; *Hong et al., 2013*). Consistent with previous findings (*Dieris et al., 2017*), cadaverine broadly activated OB mitral neurons in the dorsal glomerulus (dG), dorso-lateral glomerulus (dlG), medio-anterior glomerulus (maG), medio-dorsal glomerulus (mdG), lateral glomerulus (lG). In addition, we observed a threefold increase in the number of *fos*-expressing neurons in the right dHb following exposure of adult zebrafish to cadaverine relative to delivery of water alone (15.73 ± 1.25 vs 5.61 ± 1.07 cells, *Figure 2E and F*). The position of the *fos*-expressing cells in the right dHb corresponded to that of the *lratd2a*-expressing neurons (11.00% ± 1.07 of *lratd2a*-expressing region; *Figure 2F'*), supporting that the *lratd2a* subpopulation responds to cadaverine in both larvae and adults.

Exposure to alarm substance prepared from adult zebrafish increased the number of *fos*-expressing cells in the lateral glomerulus (lG) and dlG of the OB as would be expected (*Mathuru et al., 2012*; *Yoshihara, 2014*), but also in the dorso-lateral region of the dHb (*Figure 2G and G'*), where transcripts co-localized to *lratd2a*-expressing neurons. In contrast to cadaverine, alarm substance activated neurons equally in both the left and right dHb (20.72 ± 2.70 cells on the left and 20.31 ± 2.53 on the right; *Figure 2H*).

## Synaptic inhibition of right dHb *lratd2a* neurons reduces aversive response to cadaverine

To confirm that *lratd2a* expressing dHb neurons play a role in processing of aversive olfactory cues, we inhibited synaptic transmission in these cells and tested adults for their reaction to cadaverine. The odorant was introduced at one end of a test tank and the time individuals spent within or outside of this region of the tank was measured. We used a preference index that is based on the position of an individual fish at a given time relative to the application site of the odorant (refer to Materials and methods).

The *Tg(lratd2a:QF2)* driver line is expected to inhibit *lratd2a*-expressing neurons in both the vHb as well as in the right dHb. We therefore devised an intersectional strategy to block the activity of neurons selectively in the right dHb, which combines Cre/lox-mediated recombination (*Förster et al., 2017*; *Satou et al., 2013*; *Tabor et al., 2019*) and the QF2/QUAS system (*Ghosh and Halpern, 2016*; *Subedi et al., 2014*). We produced transgenic fish expressing Cre recombinase under the control of the endogenous *solute carrier family 5 member 7* a (*slc5a7a*) gene using CRISPR/Cas9 targeted integration (*Kimura et al., 2014*; *Figure 3A–B*). *slc5a7a* encodes a choline transporter involved in acetylcholine biosynthesis and, in zebrafish larvae, is strongly expressed in the right dHb and not in the vHb (*Hong et al., 2013*). Accordingly, in larvae bearing the three transgenes *Tg(lratd2a:QF2)[c601]*, *Tg(slc5a7a:Cre)[c662]* and *Tg(QUAS:loxP-mCherry-loxP-GFP-CAAX)[c679]*, Cre-mediated recombination resulted in a switch in reporter labeling from red to green in right dHb neurons (*Figure 3C*). We followed a similar approach to inhibit synaptic transmission from *lratd2a* right dHb neurons using Botulinum neurotoxin (*Lal et al., 2018*; *Sternberg et al., 2016*; *Zhang et al., 2017*). A *BoTxBLC-GFP* fusion protein was placed downstream of a floxed *mCherry* reporter to generate *Tg(QUAS:loxP-mCherry-loxP-BoTxBLC-GFP)[c674]*. To validate the effectiveness of this transgenic line, a neuron specific promoter from a *Xenopus* neural-specific beta tubulin (*Xla.Tubb2*) gene was used to drive QF2 expression. Larvae bearing *Tg(Xla.Tubb2:QF2;he1.1:mCherry)[c663]*; *Tg(slc5a7a:Cre)[c662]* and *Tg(QUAS:loxP-mCherry-loxP-BoTxBLC-GFP)[c674]* showed a significantly reduced response to a touch

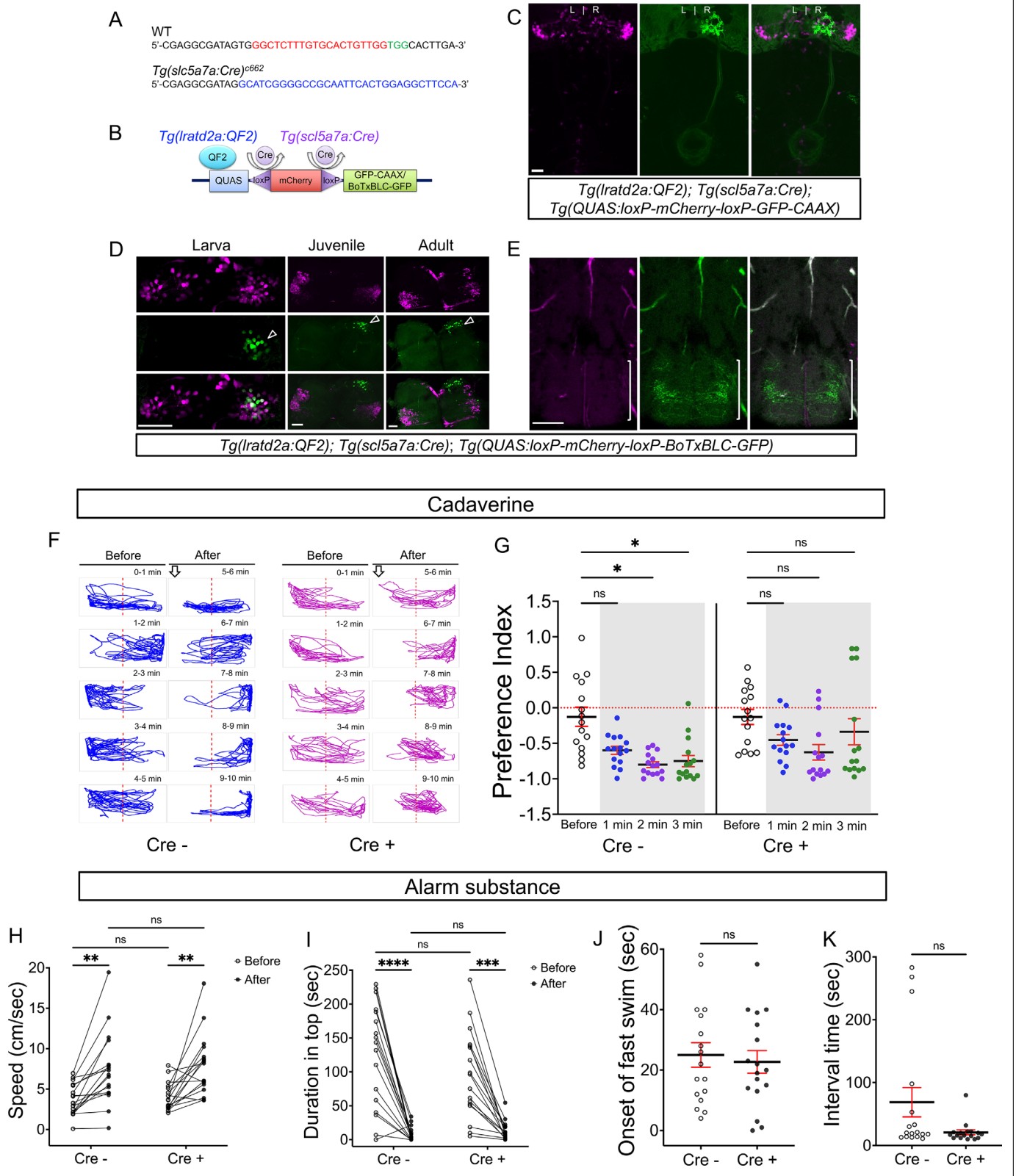

**Figure 3.** Synaptic inhibition of *lratd2a* right dHb neurons attenuates response to cadaverine. (**A**) Sequences upstream of the *slc5a7a* transcriptional start site before (WT) and after integration of Cre (blue indicates donor DNA) at sgRNA target site (red nucleotides and PAM sequences in green). (**B**) Schematic diagram of intersectional strategy using Cre/lox mediated recombination and the QF2/QUAS binary system. QF2 is driven by *lratd2a* regulatory sequences and the *slc5a7a* promoter drives Cre leading to reporter/effector expression in *lratd2a* neurons in the right dHb. (**C**) Dorsal

*Figure 3 continued on next page*

*Figure 3 continued*

view of GFP labeling in only the right dHb after Cre-mediated recombination in a 5 dpf *Tg(lratd2a:QF2)*, *Tg(slc5a7a:Cre)*, *Tg(QUAS:loxP-mCherry-loxP-GFP-CAAX)* larva. Scale bar, 25 µm. (**D**) *BoTxBLC-GFP*-labeled cells (open arrowhead) in the right dHb in *Tg(lratd2a:QF2)*, *Tg(slc5a7a:Cre)*, *Tg(QUAS:loxP-mCherry-loxP-BoTxBLC-GFP)* 5 dpf, 37 dpf, and 4 mpf zebrafish. Upper images show mCherry-labeled *lratd2a* Hb neurons, middle images show the subset of right dHb neurons that switched to GFP expression, and the bottom row are merged images. Scale bar, 50 µm. (**E**) Transverse section of *BoTxBLC-GFP* labeled axonal endings of dHb neurons that express Cre and *lratd2a* in a subregion of the vIPN (bracket) in *Tg(lratd2a:QF2)*, *Tg(slc5a7a:Cre)*, *Tg(QUAS:loxP-mCherry-loxP-BoTxBLC-GFP)* 37 dpf juveniles. Scale bar, 50 µm. (**F, G**) Preferred tank location prior to and after cadaverine addition of adults genotyped for absence (Cre-) or presence (Cre+) of *Tg(slc5a7a:Cre)*. (**F**) Representative 1 min traces for single Cre- (blue) and Cre+ (purple) adults recorded over 10 min prior to (min 0–5) and after (min 6–10) addition of cadaverine to one end of the test tank (open arrows). (**G**) Preference index for all adults for an average of 2 min before (white) and for each of 3 min after (gray) the addition of cadaverine. In Cre- fish, aversive behavior was significantly increased at 2 min (p = 0.0116) and 3 min (p = 0.0344), n = 15 fish for each group. In contrast, Cre+ fish, showed no significant difference in their preferred location over time. Dashed red lines in F and G denote midpoint of test tank. Two-way ANOVA reveals significant effects of time [$F_{(3, 27)}$ = 29, p < 0.0001], but no effect of group [$F_{(1, 14)}$ = 2.381] and interaction [$F_{(3, 33)}$ = 1.813]. Post-hoc analysis by Bonferroni's multiple comparisons. (**H**) Swimming speed during 1 min period before and after addition of alarm substance was similar for Cre- [3.68 ± 0.47 and 7.43 ± 1.1 cm/s] and Cre+ [4.02 ± 0.42 s and 7.93 ± 0.92 cm/s] adults. Two-way ANOVA reveals significant effects of time [$F_{(1, 16)}$ = 39.61, p < 0.0001], but no effect of group [$F_{(1,16)}$ = 0.2236] and interaction [$F_{(1,16)}$ = 0.0141]. Post-hoc analysis by Bonferroni's multiple comparisons. (**I**) Duration in the upper half of the test tank prior to and after addition of alarm substance for Cre+ adults was 96.6 ± 15.72 s and 13.23 ± 3.34 s and for Cre- adults was 125.53 ± 18.6 s and 8.26 ± 2.5 s. Two-way ANOVA reveals a significant effect of time [$F_{(1, 16)}$ = 63.79, p < 0.0001], but no effect of group [$F_{(1,16)}$ = 1.048] and interaction [$F_{(1,16)}$ = 0.0141]. Post-hoc analysis by Bonferroni's multiple comparisons. (**J**) Onset of fast swimming after application of alarm substance was observed at 25 ± 4.05 and at 22.7 ± 3.72 sec for Cre- and Cre+ fish, respectively [p = 0.679, unpaired *t*-test]. (**K**) Time interval between increased swimming speed and freezing behavior for Cre- (68.88 ± 23.36 s) and Cre+ (20.88 ± 3.93 s) adults [p = 0.051, unpaired *t*-test]. For H-K, all numbers represent the mean ± SEM.

The online version of this article includes the following source data and figure supplement(s) for figure 3:

**Source data 1.** Source data associated with *Figure 3G–K*.

**Figure supplement 1.** Preferred tank location prior to and after cadaverine addition within each group and between groups.

**Figure supplement 1—source data 1.** Source data associated with *Figure 3—figure supplement 1A*.

**Figure supplement 2.** Validation of intersectional strategy to inhibit cholinergic neurons using botulinum neurotoxin.

**Figure supplement 3.** Variability in *BoTxBLC-GFP* labeling of dHb neurons.

**Figure supplement 3—source data 1.** Source data associated with *Figure 3—figure supplement 3B*.

**Figure supplement 4.** Aversive response to alarm substance is intact in *BoTxBLC-GFP* juvenile fish.

---

stimulus, indicating that the neurotoxin was produced in the presence of Cre recombinase (*Figure 3—figure supplement 2*, *Video 1*).

*BoTxBLC-GFP* was selectively expressed in *lratd2a/slc5a7a* neurons of the right dHb (*Figure 3D*) in individuals bearing the three transgenes *Tg(lratd2a:QF2)*[c601], *Tg(slc5a7a:Cre)*[c662], and *Tg(QUAS:loxP-mCherry-loxP-BoTxBLC-GFP)*[c674]. Axons labeled by *BoTxBLC-GFP* terminated at the vIPN (*Figure 3E*), in the same location as those observed in *Tg(lratd2a:QF2)*, *Tg(QUAS:mApple-CAAX)* fish (*Figure 1G*) suggesting that botulinum neurotoxin inhibits synaptic transmission within this restricted region of the vIPN. We confirmed that *BoTx-BLC-GFP* labeling persisted in the right dHb neurons throughout development, although some variability was observed in the number of *BoTxBLC-GFP*-positive cells between individuals (*Figure 3—figure supplement 3*).

To determine whether the *lratd2a* neurons in the right dHb contributed to the aversive response to cadaverine, we monitored the behavior within and between groups of adults that had or did not have the *BoTxBLC-GFP* transgene. Fish lacking the transgene showed a significantly reduced preference for the side of the tank where cadaverine had been applied (*Figure 3G*).

When the response of individual fish within each group was compared over time (*Figure 3—figure*

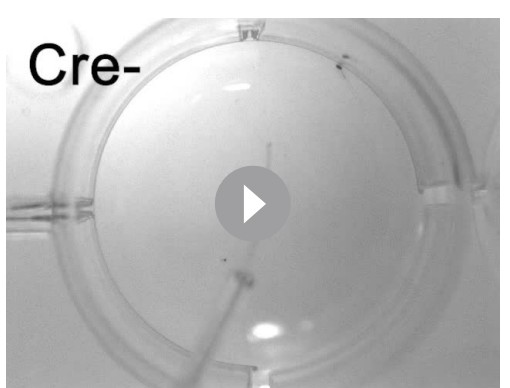

**Video 1.** Behavior of *Tg(Xla.Tubb:QF2)*, *Tg(QUAS:loxP-mCherry-loxP-BoTxBLC-GFP)* 4 dpf larvae with or without the *slc5a7a:Cre* transgene in response to touch stimulus.

https://elifesciences.org/articles/72345/figures#video1

*supplement 1*), adults with or without *BoTxBLC-GFP* initially avoided the side of the test tank where cadaverine had been introduced. However, aversion was sustained for 4 min in control fish, but not in those expressing *BoTxBLC-GFP* in *lratd2a* neurons. These findings, from statistical tests on individuals both within and between groups, suggest that *lratd2a* neurons in the right dHb are required for a prolonged aversive response to cadaverine.

Disruption of synaptic transmission in *lratd2a*-expressing Hb neurons alone did not alter the response of zebrafish to alarm substance, which typically triggers erratic, rapid swimming and bottom dwelling, followed by freezing behavior (*Diaz-Verdugo et al., 2019*; *Jesuthasan and Mathuru, 2008*). Similar to controls, both juveniles and adults expressing *BoTxBLC-GFP* under the control of *Tg(lratd2a:QF2)*[c601] showed rapid swimming/darting behavior within 22–25 s after delivery of alarm substance, first doubling their speed of swimming (*Figure 3H–K* and *Figure 3—figure supplement 4*), and then freezing for the duration of the 5 min recording period. Blocking the activity of *lratd2a* neurons in the right dHb is therefore insufficient to diminish the robust behavioral changes elicited by alarm substance (*Figure 3H–K* and *Figure 3—figure supplement 4*).

## Zebrafish mutants with habenular defects show altered responses to aversive cues

We examined the response to aversive odorants by *tcf7l2*[zf55] mutants that develop with symmetric left-isomerized dHb and lack the vHb, but are viable to adulthood (*Hüsken et al., 2014*; *Muncan et al., 2007*). In agreement with the transformation of dHb identity, projections from OB mitral cells do not terminate in the right dHb of homozygous mutants nor are *lratd2a*-expressing neurons or their efferents to the vIPN detected (*Figure 4A–D*).

Following application of cadaverine, *tcf7l2*[zf55] homozygous adults failed to exhibit the characteristic aversive behavior of their wild-type siblings (*Figure 4E and F* and *Figure 4—figure supplement 1*). Exposure to alarm substance also did not elicit a significant increase in swimming speed from baseline (1.13 ± 0.22 cm/s before and 1.89 ± 0.56 cm/s after) relative to WT siblings (2.84 ± 0.48 cm/s before and 4.88 ± 0.63 cm/s after, *Figure 4G, I and J*). Homozygous *tcf7l2*[zf55] mutants tended to swim more slowly (*Figure 4G*) and spend more time in the top half of a novel test tank than wild-type adults, although the latter behavior was suppressed in the presence of alarm substance (*Figure 4H*). *tcf7l2*[zf55] mutants are thus partly responsive to this fearful cue.

To further assess the role of *lratd2a*-expressing neurons in aversive olfactory processing, we looked at homozygous mutants of the *brain-specific homeobox* (*bsx*) gene, which develop right-isomerized dHb [(*Schredelseker and Driever, 2018*) and *Figure 5A*] and are viable to adulthood (*Schredelseker and Driever, 2018*). As might be expected when both dHb have right identity, equivalent populations of *lratd2a*-expressing neurons were found on both sides of the brain (*Figure 5C*). Instead of innervating only the right dHb as in controls, the axons of *lhx2a:gap-YFP*-labeled olfactory mitral cells terminated in the left and right dHb [(*Dreosti et al., 2014*) and *Figure 5B*], where the clusters of *lratd2a* neurons are situated (data not shown). Projections from the *lratd2a* dHb neurons coursed bilaterally through the left and right fasciculus retroflexus (FR) and innervated the same limited region of the ventral IPN (*Figure 5C*).

To measure the reaction to cadaverine in *bsx*[m1376/m1376] adults with bilaterally symmetric *lratd2a* neurons, we counted the number of cells expressing *fos* in the dHb and found an increase in the left nucleus compared to heterozygous siblings (*Figure 5D–E*). Despite the symmetric activation of dHb neurons, *bsx* homozygotes and heterozygotes both showed reduced responsiveness to cadaverine (*Figure 5F* and *Figure 5—figure supplement 1*). Overall, homozygous mutants were slower swimmers than heterozygotes (*Figure 5G–J*); however, after exposure to alarm substance, their swimming speed relative to baseline was twofold faster than that of their heterozygous siblings (*Figure 5G*), indicative of an enhanced response to this aversive cue.

## Discussion

From worms to humans, stimuli including odors are differently perceived by left and right sensory organs to elicit distinct responses (*Güntürkün and Ocklenburg, 2017*; *Güntürkün et al., 2020*). Honeybees, for example, show an enhanced performance in olfactory learning when their right antenna is trained to odors (*Guo et al., 2016*; *Letzkus et al., 2006*; *Rogers and Vallortigara, 2008*).

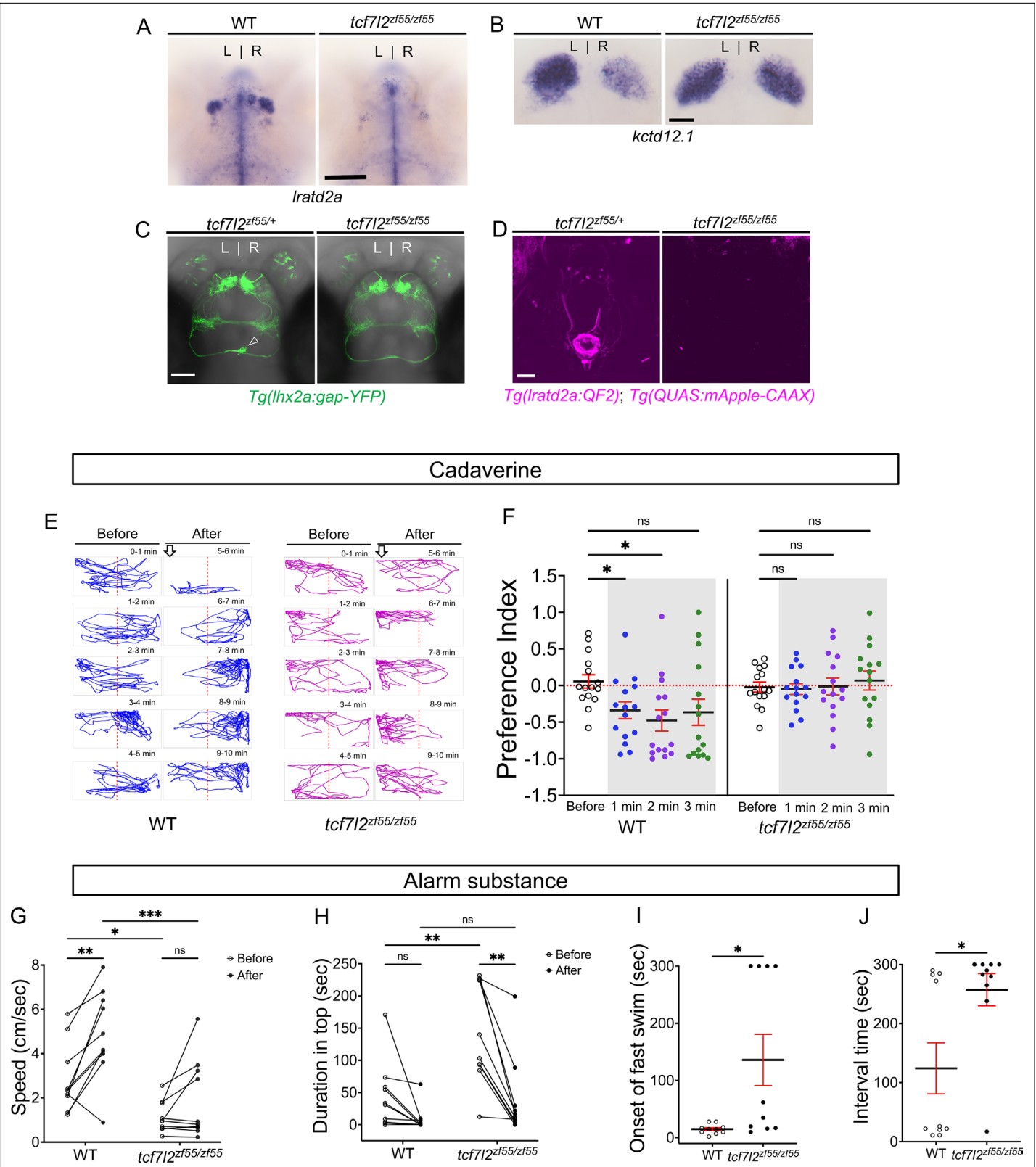

**Figure 4.** Attenuated response to aversive odorants by left-isomerized dHb mutants. (**A–B**) (**A**) Absence of *lratd2a*-expressing right dHb neurons and (**B**) right-isomerized expression of *kctd12.1* in *tcf7l2* mutant larvae at five dpf. (**C**) Dorsal views of olfactory mitral neuronal projections of *Tg(lhx2a:gap-YFP)* larvae at 6 dpf. Open arrowhead indicates axon terminals of mitral cells in the WT right dHb that are absent in the mutant. (**D**) Dorsal views of dHb neuronal projections to the ventral IPN in *Tg(lratd2a:QF2), Tg(QUAS:mApple-CAAX)* larvae at 6 dpf. (**E**) Representative traces (1 min) for *tcf7l2* mutant

*Figure 4 continued on next page*

*Figure 4 continued*

and WT sibling adults after application of cadaverine. (**F**) Preference index for mutants and WT siblings for an average of 2 min before (white) and for each of 3 min after (gray) the addition of cadaverine. Only WT fish showed a significant difference in their preferred location at 1 min (p = 0.0439) and at 2 min (p = 0.0184). For each group, n = 15 adults. Two-way ANOVA reveals a significant effect of time [$F_{(3, 24)}$ = 3.665, p = 0.046], group [$F_{(1, 14)}$ = 6.197, p = 0.026] and interaction [$F_{(3, 30)}$ = 7.953, p = 0.001]. Post-hoc analysis by Bonferroni's multiple comparisons. Dashed red lines denote midpoint of test tank. (**G**) Swimming speed for 30 s before and after addition of alarm substance was1.13 ± 0.22 cm/s and 1.89 ± 0.56 cm/s for *tcf7l2* homozygotes and 2.84 ± 0.48 cm/s and 4.88 ± 0.63 cm/s for their WT siblings, n = 10 fish for each group. Two-way ANOVA reveals significant effects of time [$F_{(1, 9)}$ = 19.31, p = 0.0021] and group [$F_{(1, 9)}$ = 13.91, p = 0.0047], but no effect of interaction [$F_{(1, 9)}$ = 3.933]. Post-hoc analysis by Bonferroni's multiple comparisons. (**H**) Duration in the upper half of the test tank prior to and after addition of alarm substance for *tcf7l2* adults was 143.58 ± 24.80 s and 38.77 ± 19.56 s and 43.68 ± 16.35 s and 8.19 ± 6.16 s for their WT siblings, n = 10 fish for each group. Two-way ANOVA reveals significant effects of time [$F_{(1, 9)}$ = 3755, p = 0.0002], group [$F_{(1, 9)}$ = 12.42, p = 0.0065] and interaction [$F_{(1, 9)}$ = 5.877, p = 0.0383]. Post-hoc analysis by Bonferroni's multiple comparisons. (**I**) Onset of fast swimming after application of alarm substance occurred at 15 ± 2.65 s for WT and at 136 ± 44.86 s for *tcf7l2* fish [p = 0.015, unpaired *t*-test]. (**J**) The time interval between increased swimming speed and freezing behavior was 124.2 ± 43.13 s for WT and 257.3 ± 27.43 s for *tcf7l2* fish [p = 0.018, unpaired *t*-test]. For F-J, all numbers represent the mean ± SEM.

The online version of this article includes the following source data and figure supplement(s) for figure 4:

**Source data 1.** Source data associated with *Figure 4E–J*.

**Figure supplement 1.** Preferred tank location prior to and after cadaverine addition within each group and between groups.

**Figure supplement 1—source data 1.** Source data associated with *Figure 4—figure supplement 1A*.

In mice, over one third of mitral/tufted cells were found to be interconnected between the ipsilateral and contralateral olfactory bulbs for sharing of odor information received separately from each nostril, and for coordinated perception (*Grobman et al., 2018*). The zebrafish provides a notable example of a lateralized olfactory pathway, with the discovery of a subset of bilateral mitral cells that project to the dorsal habenulae but terminate only at the right nucleus (*Miyasaka et al., 2014*; *Miyasaka et al., 2009*). This finding prompted us to ask what is different about the post-synaptic dHb neurons that receive this olfactory input and what function does this asymmetric pathway serve.

## Aversive olfactory cues activate identified neurons in the right dHb

We previously showed that the olfactory mitral cells that express *lhx2a* and are located in medio-dorsal and ventro-medial bilateral glomerular clusters (*Miyasaka et al., 2014*; *Miyasaka et al., 2009*) and project their axons to a subregion of the right dHb where the *lratd2a* gene is transcribed (*deCarvalho et al., 2013*). From transgenic labeling with membrane-tagged fluorescent proteins and synaptophysin, we now confirm that the *lhx2a* olfactory neurons precisely terminate at a cluster of *lratd2a/slc5a7a* expressing cholinergic neurons present in the right dHb.

Through calcium imaging using a genetically encoded calcium indicator, we validated that the right dHb is responsive when larval zebrafish are exposed to aversive odors such as cadaverine or chondroitin sulfate (*Jetti et al., 2014*; *Krishnan et al., 2014*), a component of alarm substance (*Mathuru et al., 2012*), more specifically, that the *lratd2a*-expressing neurons of the right dHb significantly respond to these aversive olfactory cues above their response to water alone. As has also been observed by others (*Jesuthasan et al., 2021*), application of vehicle alone, even when introduced slowly into a testing chamber, is sufficient to elicit a change in GCaMP fluorescence.

In adults, we used *fos* expression as a measure of neuronal activation and showed that transcripts colocalized to *lratd2a*-expressing cells. Interestingly, cadaverine predominantly activated neurons in the right dHb in larvae and adults, whereas neurons responsive to alarm substance were detected in both the left and right dHb nuclei of adult zebrafish. Different types of olfactory cues activate distinct glomeruli in the OB (*Friedrich and Korsching, 1997*; *Yoshihara, 2014*), and consistent with the prior studies, we observed that, in adults, cadaverine significantly increased *fos* expression in the mdG and dG regions, the location of *lhx2a* neurons that project to the right dHb. By contrast, alarm substance predominantly activated neurons in the lG and dlG regions of the OB that innervate the telencephalon and posterior tuberculum (*Miyasaka et al., 2014*; *Miyasaka et al., 2009*), suggesting that both dHb receive input via this route rather than through direct olfactory connections. Indeed, we found that more neurons reacted to alarm substance than cadaverine throughout the brain, including in the Dp, Vv and thalamic areas (data not shown).

In previous experiments (*deCarvalho et al., 2013*), we did not detect activated neurons in the right dHb of adult zebrafish following exposure to cadaverine or alarm substance. Several factors could

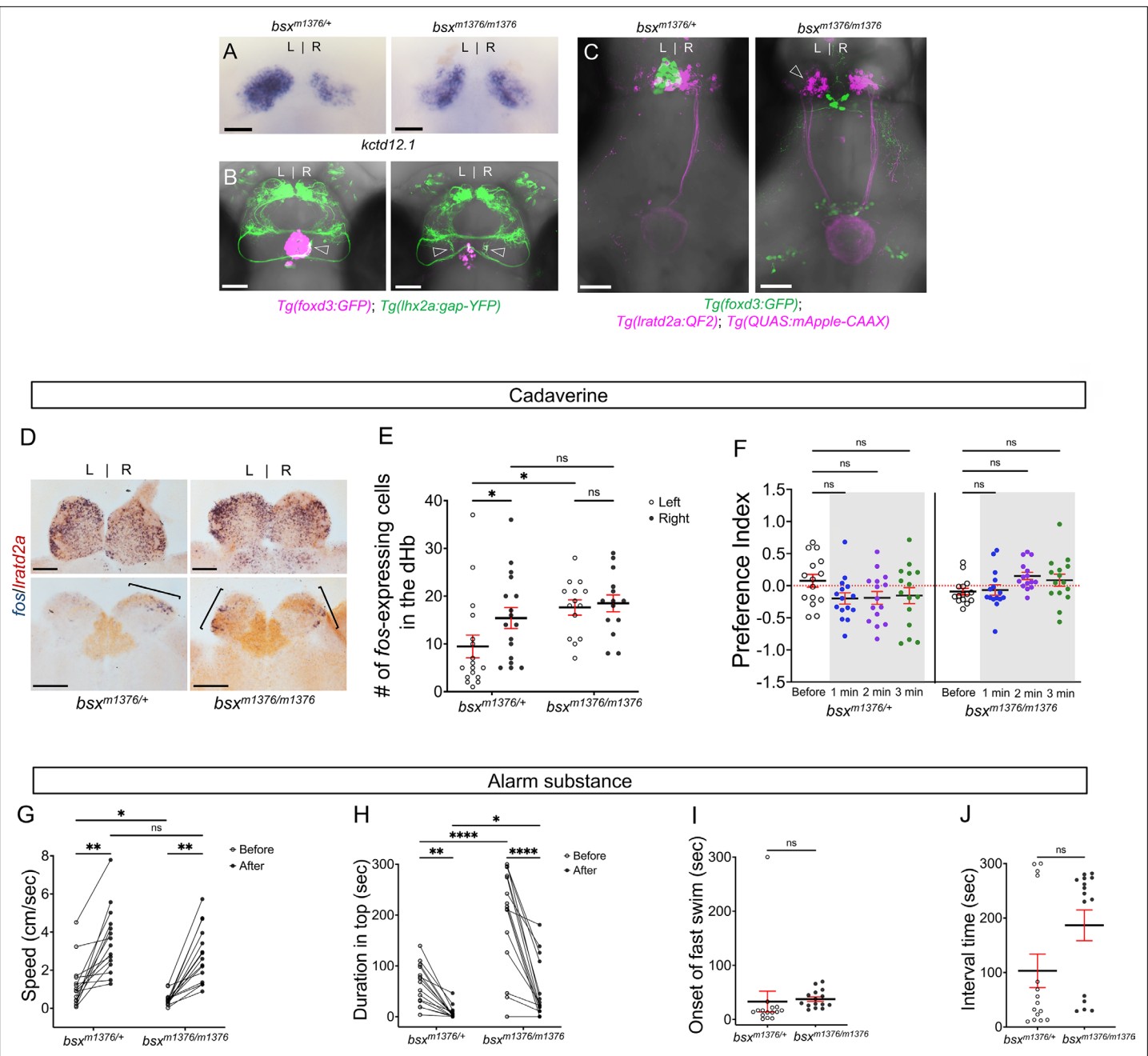

**Figure 5.** Enhanced reactivity to alarm substance in mutants with right-isomerized dHb. (**A**) Asymmetric expression pattern of *kctd12.1* is right-isomerized in *bsx* homozygotes at five dpf. (**B**) Projections of *Tg(lhx2a:gap-YFP)* labeled olfactory mitral cells terminate bilaterally (open arrowheads) in the dHb of *bsx^m1376^* homozygous mutants at five dpf. (**C**) In *bsx* mutants, axons from both left (open arrowhead) and right dHb *lratd2a* neurons project to the same region of the vIPN. Scale bar, 50 μm. (**D**) Bilateral *fos*-expressing neurons in right-isomerized mutants. *fos* (blue) and *lratd2a* (brown) transcripts in the olfactory bulbs (upper panels) and dHb (bottom panels) of 10-month-old *bsx^m1376^* heterozygotes and homozygous mutants detected by RNA in situ hybridization 30 min after addition of cadaverine to the test tank. Brackets indicate *fos*-expressing cells. Scale bar, 100 μm. (**E**) Quantification of *fos*-expressing cells in the dHb after application of cadaverine in *bsx^m1376/+^* [9.47 ± 2.37 cells on the left and 15.41 ± 2.19 cells on the right, n = 17 sections from nine adults] and *bsx^m1376/m1376^* adults [17.64 ± 1.59 cells on the left and 18.5 ± 1.76 cells on the right, n = 14 sections from eight adults]. Two-way mixed ANOVA reveals significant effect of group [$F_{(1, 16)}$ = 5.178, p = 0.037] and left vs. right [$F_{(1, 16)}$ = 6.885, p = 0.0184], but no effect of interaction [$F_{(1, 10)}$ = 3.85]. Post-hoc analysis by Bonferroni's multiple comparisons. (**F**) Preference index for *bsx* adults for an average of 2 min before (white) and for each of 3 min after (gray) the addition of cadaverine. Both *bsx* homozygotes and heterozygotes showed reduced responsiveness to cadaverine. Two-way ANOVA reveals significant effect of interaction [$F_{(3, 34)}$ = 5.483, p = 0.005], but no effect of time [$F_{(3, 25)}$ = 0.987] and group [$F_{(1, 14)}$ = 2.728]. Post-hoc analysis by Bonferroni's multiple comparisons. (**G**) Swimming speed for 30 s before and after addition of alarm substance. In heterozygous adults, swimming speed was 1.22 ± 0.31 cm/s before and 3.52 ± 0.44 cm/sec after and, in homozygotes, 0.46 ± 0.08 cm/s before and 2.80 ± 0.37 cm/s after, n = 15 adults

*Figure 5 continued on next page*

*Figure 5 continued*

for each group. Two-way ANOVA reveals significant effect of time [$F_{(1, 14)}$ = 113.4, p < 0.0001], but no effect of group [$F_{(1, 14)}$ = 4.459] and interaction [$F_{(1, 14)}$ = 0.0023]. Post-hoc analysis by Bonferroni's multiple comparisons. (**H**) Duration in the upper half of the test tank prior to and after addition of alarm substance for *bsx*$^{m1376/m1376}$ adults was 194.86 ± 25.66 s and 51.89 ± 14.84 s and was 63.55 ± 10.11 s and 7.95 ± 3.24 s for *bsx*$^{m1376/+}$, n = 15 fish for each group. Two-way ANOVA reveals significant effect of time [$F_{(1, 14)}$ = 44.35, p < 0.0001], group [$F_{(1, 14)}$ = 22.45, p = 0.0003] and interaction [$F_{(1, 14)}$ = 20.89, p = 0.0004]. Post-hoc analysis by Bonferroni's multiple comparisons. (**I**) Onset of fast swimming after application of alarm substance was observed at 33.13 ± 19.19 s in *bsx*$^{m1376/+}$ and at 37.73 ± 4.27 s in *bsx*$^{m1376/m1376}$ fish [p = 0.816, unpaired *t*-test]. (**J**) Time interval between increased swimming speed and freezing behavior for *bsx*$^{m1376/+}$ (103.1 ± 30.75 s) and for *bsx*$^{m1376/m1376}$ (186.7 ± 28.23 s) [p = 0.055, unpaired *t*-test]. For E-J, all numbers represent the mean ± SEM.

The online version of this article includes the following source data and figure supplement(s) for figure 5:

**Source data 1.** Source data associated with *Figure 5E–J*.

**Figure supplement 1.** Preferred tank location prior to and after cadaverine addition within each group and between groups.

**Figure supplement 1—source data 1.** Source data associated with *Figure 5—figure supplement 1A*.

account for the difference from the earlier study: we now have the transgenic tools to examine *lratd2a* neurons directly, we used higher concentrations of cadaverine and alarm substance and, in contrast to delivering odorants to groups of zebrafish, we tested the neuronal response in individual adults.

It has been suggested that lateralized olfactory and visual functions of the dHb are more prominent early in development and less so at later stages (*Fore et al., 2020*). However, the presence of *lratd2a*-expressing neurons in the right dHb and their preferential response to cadaverine from larval to adult stages supports the persistence of lateralized activity and illustrates the value of examining defined neuronal populations.

## Right dHb neurons mediate aversive behavioral responses

As a group, zebrafish in which the synaptic activity of *lratd2a* neurons in the right dHb was inhibited by *BoTxBLC-GFP* did not exhibit repulsion to cadaverine, although some individuals showed a mild aversive response that was not sustained relative to sibling controls. This subtle behavioral effect suggests that the entire population of *lratd2a* neurons may not be effectively inactivated in all animals. The described intersectional approach to suppress neuronal activity may be incomplete, as evidenced by the observed variability in *BoTxBLC-GFP* labeling between individual larvae. Additionally, compensatory mechanisms occurring between the onset of Cre expression in *slc5a7a* neurons (at 3 dpf) and adulthood could allow for partial recovery of the response to repulsive cues. It is also possible that the *lratd2a* neurons are a heterogeneous population, with different subsets controlling the magnitude of the aversive response or the duration. More finely tuned techniques for temporal or spatial regulation of neuronal inactivation would help resolve these issues.

Despite both being aversive cues (*Hussain et al., 2013*; *Mathuru et al., 2012*), cadaverine and alarm substance elicit different behavioral responses by adult zebrafish. Control fish show active repulsion to cadaverine for the first to 2–4 min of a 5 min testing period, whereas alarm substance triggers immediate erratic behavior such as rapid swimming and darting that is typically followed by prolonged freezing (*Hussain et al., 2013*; *Mathuru et al., 2012*). Therefore, it is not necessarily expected that the same neuronal populations will mediate the response to both substances.

Perturbation of the *lratd2a*-expressing right dHb neurons either selectively by *BoTxBLC*-mediated synaptic inactivation, or in *tcf7l2*$^{zf55}$ homozygous mutants that completely lack them, reduced aversion to cadaverine, either in the length or degree of the response. In contrast to juveniles or adults with *BoTxBLC* inactivated neurons that displayed a similar response to alarm substance as controls, *tcf7l2*$^{zf55}$ mutants, showed no difference in their swimming behavior before and after its addition. One explanation is that many regions throughout the brain are likely involved in directing the complex repertoire of behaviors elicited by alarm substance and inactivation of *lratd2a* neurons in the habenular region alone is insufficient to weaken the overall response. Furthermore, the *tcf7l2*$^{zf55}$ mutation could disrupt other brain regions that regulate behaviors elicited by alarm substance since the *tfc7l2* gene is expressed in neurons throughout the brain, including the anterior tectum, dorsal thalamus and the hindbrain (*Young et al., 2002*).

Similar to *tcf7l2*$^{zf55}$, the *bsx*$^{m1376}$ mutation is pleiotropic resulting in right-isomerization of the dHb due to the absence of the parapineal (*Schredelseker and Driever, 2018*), and also the loss of the terminal tuberal hypothalamus, mammillary hypothalamic regions and secondary prosencephalon

(*Schredelseker et al., 2020*). We did not observe enhanced or prolonged aversion to cadaverine in *bsx^m1376* homozygotes relative to controls. However, although homozygous mutants did show a hyperactive response to alarm substance, we cannot discount the involvement of other affected brain regions. Albeit technically challenging in adults, a more selective test such as optogenetic activation of only the *lratd2a* dHb neurons in wild-type and mutant zebrafish could help resolve their contribution to the alarm response.

The identification of a subset of neurons in the right dHb that receive olfactory input and terminate their axons at a defined subregion of the ventral IPN lays the groundwork for tracing an entire pathway from olfactory receptors to the neurons directing the appropriate behavioral response. The midline IPN has been morphologically defined into subregions (*deCarvalho et al., 2014* ; *Lima et al., 2017*; *Quina et al., 2017*), but their connectivity and functional properties are not well studied. Recent work has begun to assign different functions to given subregions, such as the role of the rostral IPN in nicotine aversion (*Morton et al., 2018*; *Quina et al., 2017*). Neurons in the ventral IPN project to the raphe nucleus (*Agetsuma et al., 2010*; *Lima et al., 2017*), but the precise identity of raphe neurons that are innervated by the *lratd2a*-expressing dHb neurons remains to be determined. Transcriptional profiling of the IPN should yield useful information on its diverse neuronal populations and likely lead to the identification of the relevant post-synaptic targets in the ventral IPN and their efferent connections. Elaboration of this pathway may also help explain the advantage of lateralization in the processing of aversive information. It has been argued, for instance, that the antennal specialization to aversive odors in bees is correlated with directed turning away from the stimulus and escape (*Rogers and Vallortigara, 2019*). Directional turning has also been observed in larval zebrafish (*Horstick et al., 2020*), but whether it is correlated with laterality of the Hb-IPN pathway is unclear. Beyond olfaction, left-right asymmetry appears to be a more general feature of stress-inducing, aversive responses as demonstrated for the rat ventral hippocampus (*Sakaguchi and Sakurai, 2017*) and human pre-frontal cortex, where heightened anxiety also activates more neurons on the right than on the left (*Avram et al., 2010*; *Ocklenburg et al., 2016*).

# Materials and methods

**Key resources table**

| Reagent type (species) or resource | Designation | Source or reference | Identifiers | Additional information |
|---|---|---|---|---|
| Genetic reagent (*Danio rerio*) | Tg(lratd2a:QF2)^c601 | This paper | | Transgenic, Halpern lab |
| Genetic reagent (*Danio rerio*) | Tg(slc5a7a:Cre)^c662 | This paper | | Transgenic, Halpern lab |
| Genetic reagent (*Danio rerio*) | Tg(Xla.Tubb2:QF2; he1.1:mCherry)^c663 | This paper | | Transgenic, Halpern lab |
| Genetic reagent (*Danio rerio*) | Tg(QUAS:GCaMP6f)^c587 | This paper | | Transgenic, Halpern lab |
| Genetic reagent (*Danio rerio*) | Tg(QUAS:GFP)^c403 | *Subedi et al., 2014* | | Transgenic, Halpern lab |
| Genetic reagent (*Danio rerio*) | Tg(QUAS:mApple-CAAX;he1.1:mCherry)^c636 | This paper | | Transgenic, Halpern lab |
| Genetic reagent (*Danio rerio*) | Tg(QUAS:loxP-mCherry-loxP-GFP-CAAX)^c679 | This paper | | Transgenic, Halpern lab |
| Genetic reagent (*Danio rerio*) | Tg(QUAS:loxP-mCherry-loxP-BoTxBLC-GFP)^c674 | This paper | | Transgenic, Halpern lab |
| Genetic reagent (*Danio rerio*) | Tg(–10lhx2a:gap-EYFP)^zf177 | *Miyasaka et al., 2009* | RRID:ZFIN_ZDB-GENO-100504-13 | Transgenic |
| Genetic reagent (*Danio rerio*) | Tg(lhx2a:syp-GFP)^zf186 | *Miyasaka et al., 2009* | RRID:ZFIN_ZDB-GENO-100504-13 | Transgenic |
| Genetic reagent (*Danio rerio*) | tcf7l2^zf55 | *Muncan et al., 2007* | RRID:ZFIN_ZDB-GENO-071217-3 | Mutant |

*Continued on next page*

*Continued*

| Reagent type (species) or resource | Designation | Source or reference | Identifiers | Additional information |
|---|---|---|---|---|
| Genetic reagent (*Danio rerio*) | *bsx^{m1376}* | *Schredelseker and Driever, 2018* | RRID:ZFIN_ZDB-GENO-180802-1 | Mutant |
| Chemical compound, drug | alpha-Bungarotoxin | Invitrogen | Cat# B-1601 | (1 mg/ml) |
| Chemical compound, drug | Cadaverine | Sigma-Aldrich | Cat# 33,211 | (100 µM) |
| Chemical compound, drug | Chondroitin sulfate sodium salt from shark cartilage | Sigma-Aldrich | Cat# C4384 | (100 µg/ml) |
| Chemical compound, drug | T7 Endonuclease I | NEB | Cat# M0302L | |
| Chemical compound, drug | MAXIscript T7 Transcription Kit | Invitrogen | Cat# AM1312 | |
| Chemical compound, drug | mMASSAGE mMACHINE T3 Transcription Kit | Invitrogen | Cat# AM1348 | |
| Chemical compound, drug | Gateway BP Clonase II Enzyme mix | Thermo Fisher Scientific | Cat# 11789020 | |
| Chemical compound, drug | Gateway LR Clonase II Enzyme mix | Thermo Fisher Scientific | Cat# 11791020 | |
| Chemical compound, drug | DIG RNA Labeling Mix | Roche | Cat# 11277073910 | |
| Chemical compound, drug | 5-bromo-4-chloro-3-indolyl-phosphate, 4-toluidine salt (BCIP) | Roche | Cat# 11383221001 | |
| Chemical compound, drug | 4-Nitro blue tetrazolium chloride, solution (NBT) | Roche | Cat# 11383213001 | |
| Chemical compound, drug | 2-(4-Iodophenyl)–3-(4-nitrophenyl)–5-phenyltetrazolium Chloride (INT) | FisherScientific | Cat# I00671G | |
| Antibody | Anti-Digoxigenin-AP, Fab fragments antibody (Sheep polyclonal) | Roche | Cat# 11093274910, | (1:5000) |
| Antibody | Anti- Fluorescein -AP, Fab fragments antibody (Sheep polyclonal) | Roche | Cat# 11426338910 | (1:5000) |
| Recombinant DNA reagent | Plasmid: Gbait-hs-Gal4 | *Kimura et al., 2014* | N/A | |
| Recombinant DNA reagent | Plasmid: NBeta-pEGFP-1 | Gift from Paul Krieg | N/A | |
| Recombinant DNA reagent | Plasmid: Gbait-hsp70-QF2-SV40pA | This paper | Addgene Plasmid #122,563 | Donor plasmid for integration, Halpern lab |
| Recombinant DNA reagent | Plasmid: Gbait-hsp70-Cre-SV40pA | This paper | Addgene Plasmid #122,562 | Donor plasmid for integration, Halpern lab |
| Recombinant DNA reagent | Plasmid: pDR274 | *Hwang et al., 2013* | Addgene Plasmid #42,250 | |
| Recombinant DNA reagent | Plasmid: GFP sgRNA | *Auer et al., 2014* | N/A | |
| Recombinant DNA reagent | Plasmid: pT3TS-nCas9n | *Jao et al., 2013* | #46,757 | |

*Continued on next page*

*Continued*

| Reagent type (species) or resource | Designation | Source or reference | Identifiers | Additional information |
|---|---|---|---|---|
| Recombinant DNA reagent | pGEM-T Easy Vector | Promega | Catalog# A1360 | |
| Software, Algorithm | Fiji | *Schindelin et al., 2012* | | https://imagej.net/Fiji |
| Software, Algorithm | MATLAB | The MathWorks | | https://www.mathworks.com/ |
| Software, Algorithm | Excel | Microsoft | | http://products.office.com/en-us/excel |
| Software, Algorithm | GraphPad Prism9 | GraphPad Software | | https://www.graphpad.com/ |
| Software, Algorithm | ZebraLab | Viewpoint Life Sciences | | http://www.viewpoint.fr/en/home |

## Zebrafish

Zebrafish were maintained at 27 °C under a 14:10 hr light/dark cycle in a recirculating system with dechlorinated water (system water). The AB wild-type strain (*Walker, 1998*), transgenic lines *Tg(lratd2a:QF2)^c601^*, *Tg(slc5a7a:Cre)^c662^*, *Tg(Xla.Tubb2:QF2;he1.1:mCherry)^c663^*, *Tg(QUAS:GCaMP6f)^c587^*, *Tg(QUAS:GFP)^c403^* (*Subedi et al., 2014*), *Tg(QUAS:mApple-CAAX;he1.1:mCherry)^c636^*, *Tg(QUAS:loxP-mCherry-loxP-GFP-CAAX)^c679^*, and *Tg(QUAS:loxP-mCherry-loxP-BoTxBLC-GFP)^c674^*, *Tg(–10lhx2a:gap-EYFP)^zf177^* (formally known as *Tg(lhx2a:gap-YFP)*) (*Miyasaka et al., 2009*), *Tg(lhx2a:syp-GFP)^zf186^* (*Miyasaka et al., 2009*), and mutant strains *tcf7l2^zf55^* (*Muncan et al., 2007*) and *bsx^m1376^* (*Schredelseker and Driever, 2018*) were used. For imaging, embryos and larvae were transferred to system water containing 0.003% phenylthiourea (PTU) to inhibit melanin pigmentation. All zebrafish protocols were approved by the Institutional Animal Care and Use Committee (IACUC) of the Carnegie Institution for Science [Protocol #122] or Dartmouth College [Protocol #00002253(m3a)].

## Generation of transgenic lines by *Tol2* transgenesis

The MultiSite Gateway-based construction kit (*Kwan et al., 2007*) was used to create transgenic constructs for Tol2 transposition. A 16 bp *QUAS* sequence (*Potter et al., 2010*), was cloned into the 5' entry vector (*pDONRP4-P1R*, #219 of Tol2kit v1.2) via a BP reaction (11789020, Thermo Fisher Scientific). Middle entry vectors (*pDONR221*, #218 of Tol2kit v1.2 *Kwan et al., 2007*) were generated for *QF2*, *mApple-CAAX*, *loxP-mCherry-stop-loxP* and *GCaMP6f*. Sequences corresponding to the *SV40 poly A* tail, the *SV40 poly A* tail followed by a secondary marker consisting of the zebrafish *hatching enzyme one* promoter (*Xie et al., 2012*) driving mCherry, or to *BoTxBLC-GFP* (*Lal et al., 2018*; *Sternberg et al., 2016*; *Zhang et al., 2017*) were cloned into the 3' entry vector (*pDONRP2R-P3*, #220 of Tol2kit v1.2 *Kwan et al., 2007*). Final constructs were created using an LR reaction (11791020, Thermo Fisher Scientific) into a Tol2 destination vector (*pDestTol2pA2*, #394 of the Tol2kit v1.2 *Kwan et al., 2007*; *Supplementary file 1*).

To produce Tol2 transposase mRNA, *pCS-zT2TP* was digested by *Not*I and RNA synthesized using the mMESSAGE mMACHINE SP6 Transcription Kit (AM1340, Thermo Fisher Scientific). RNA was purified by phenol/chloroform-isoamyl extraction, followed by chloroform extraction and isopropanol precipitation (*Suster et al., 2011*). A solution containing *QF2/QUAS* plasmid DNA (~25 ng/µl), transposase mRNA (~25 ng/µl) and phenol red (0.5%) was microinjected into one-cell stage zebrafish embryos, which were raised to adulthood. To identify transgenic founders, $F_0$ adult fish were outcrossed to AB and embryos were assessed for the presence of the secondary marker by screening for mCherry labeling of hatching gland cells after 24 hr post fertilization (hpf) and raised to adulthood.

## Generation of transgenic lines by genome editing

For generating transgenic lines at targeted sites, we performed CRISPR/Cas9-mediated genome editing using the method of *Kimura et al., 2014*, which relies on homology-independent repair of double-strand breaks for integration of donor DNA. To construct the donor DNA, we combined GFP bait sequences (Gbait) and the hsp70 promoter fragment (*Kimura et al., 2014*), with a QF2 sequence,

which contains the DNA binding and transcriptional activation domains of the QF transcription factor of *Neurospora crassa* (*Ghosh and Halpern, 2016*; *Subedi et al., 2014*). The Gbait-hsp70 sequence was amplified with forward 5'- GGCGAGGGCGATGCCACCTACGG-3' and reverse 5'- CCGCGGCA AGAAACTGCAATAAAAAAAAC-3' primers, using Gbait-hsp70:Gal4 donor DNA (*Kimura et al., 2014*). QF2 sequence was amplified with forward 5'- ACTAGTATGCCACCCAAGCGCAAAACGC-3' and reverse 5'- CTGCAGCAACTATGTATAATAAAGTTGAAA-3' primers, using pDEST:QF2 template DNA. Subsequently, the Gbait-hsp70 fragment and QF2 fragment were independently inserted into pGEM T-easy (A1360, Promega) and subsequently combined into one vector by *Sac*II digestion and ligation (Addgene, plasmid #122563). The Cre sequence was amplified using pCR8GW-Cre-pA-FRT-kan-FRT as template DNA (*Suster et al., 2011*) (forward 5'-ACTAGTGCCACCATGGCCAATTTACTG-3' and reverse 5'-CTGCAGGGACAAACCACAACTAGA-3' primers), and inserted into pGEM T-easy. The Gbait-hsp70 fragment was subcloned into the Cre vector by *Sac*II digestion and ligation (Addgene, plasmid #122562).

Production of sgRNAs and Cas9 RNA was performed as described previously (*Hwang et al., 2013*; *Jao et al., 2013*). Potential sgRNAs were designed using Zifit (*Sander et al., 2010*). Pairs of synthetic oligonucleotides (*lratd2a* sense, 5'-TAGGACTGGACACCGAAGAAGA-3'; *lratd2a* anti-sense, 5'-AAACTCTTCTTCGGTGTCCAGT-3'; *slc5a7a* sense, 5'-TAGGCTCTTTGTGCACTGTTGG-3'; *slc5a7a* anti-sense, 5'-AAACCCAACAGTGCACAAAGAG-3'), 5'-TAGG-N$_{18}$-3' and 5'-AAAC-N$_{18}$-3', were annealed and inserted at the *Bsa*I site of the pDR274 vector (Addgene, plasmid #42250). To make sgRNA and Cas9 mRNA, template DNA for sgRNAs and pT3TS nCas9n (Addgene, plasmid #46757) were digested by *Dra*I and *Xba*I, respectively. The MAXIscript T7 Transcription Kit (AM1312, Thermo Fisher Scientific) was used for synthesis of sgRNAs from linearized DNA template and the mMESSAGE mMACHINE T3 Transcription Kit (AM1348, Thermo Fisher Scientific) for synthesis of Cas9 RNA. RNA was purified by phenol/chloroform and precipitated by isopropanol.

A solution containing sgRNA for the targeted gene (~50 ng/μl), sgRNA (~50 ng/μl) to linearize donor plasmids at the Gbait site (*Auer et al., 2014*; *Kimura et al., 2014*), the Gbait-hsp70-QF2-pA and Gbait-hsp70-Cre-pA (~50 ng/μl) plasmids, Cas9 mRNA (~500 ng/μl), and phenol red (0.5%) was microinjected into one-cell stage embryos. To verify integration of donor DNA at the target locus, PCR was performed using primers that correspond to sequences flanking the integration site and within the donor plasmid (*hsp70* reverse, 5'-TCAAGTCGCTTCTCTTCGGT-3'). (For *lratd2a, the* forward primer is 5'-CTGCTGAAGTGGCATTTATGGGC-3' and the reverse primer is 5'-CCTGGAAGTCCCCG-ACATAC-3'; for *slc5a7a* the forward primer is 5'-CACATCTCTCTGACGTCCATC-3' and the reverse is 5'-GTTGCTGCGCAGGACTTAAAA-3'). Sequence analysis of PCR products confirmed integration at the targeted sites.

## RNA in situ hybridization

Whole-mount in situ hybridization was performed as previously described (*deCarvalho et al., 2014*; *Gamse et al., 2002*). In brief, larvae and dissected brains were fixed in 4% paraformaldehyde (P6148, Sigma-Aldrich) in 1 X PBS (phosphate-buffered saline) at 4 °C overnight. To synthesize RNA probes, the following restriction enzymes and RNA polymerases were used: *lratd2a* (*Bam*HI/T7), *fos* (*Not*I/SP6), *slc5a7a* (*Not*I/SP6), *kctd12.1* (*Eco*RI/T7) (*deCarvalho et al., 2013*; *Hong et al., 2013*). Probes were labeled with UTP-digoxigenin (11093274910, Roche) and samples incubated at 70 °C in hybridization solution containing 50% formamide. Hybridized probes were detected using alkaline phosphatase-conjugated antibodies (Anti-Digoxigenin-AP, #11093274910, and Anti-Fluorescein-AP, #11426338910, Sigma-Aldrich) and visualized by staining with 4-nitro blue tetrazolium (NBT, #11383213001, Roche), 5-bromo-4-chloro-3-indolyl-phosphate (BCIP, #11383221001, Roche) and 2-(4-Iodophenyl)–3-(4-nitro phenyl)–5-phenyltetrazolium Chloride (INT, #I00671G, Fisher Scientific).

## Preparation of odorants

Alarm substance was freshly prepared on the day of testing. Adult zebrafish (6 female and six male) were anesthetized in 0.02% tricaine (E10521, Ethyl 3-aminobenzoate methanesulfonate; Sigma-Aldrich). Shallow lesions were made on the skin (10 on each side) using a fresh razor blade and single fish were consecutively immersed in a beaker containing distilled water (25 ml for *fos* experiment and 50 ml for behavioral analyses) for 30 s at 4 °C. The solution was filtered using a 0.2 μm filter (565–0020, ThermoFisher Scientific) and stored at 4 °C until used (*Mathuru et al., 2012*). The cadaverine (#33211,

final concentration 100 µM) and chondroitin sulfate (#c4384, 100 µg/ml) were purchased from Sigma-Aldrich, and stock solutions prepared in distilled water.

## Calcium imaging in larval zebrafish

To monitor activity within the same population of *lratd2a* neurons in response to odorants or to a vehicle control, calcium signaling was performed on seven dpf *Tg(lratd2a:QF2)[c644]; Tg(QUAS:G-CaMP6f)[c587]* larvae. Larvae were paralyzed by immersion in α-bungarotoxin (20 µl of 1 mg/ml solution in system water, B1601, ThermoFisher Scientific) *Duboué et al., 2017; Severi et al., 2014* followed by washing in fresh system water. Individuals were embedded in 1.5% low melting agarose (SeaPlaque Agarose, Lonza) in a petri dish (60 mm) with a custom-designed mold. After solidification, the agarose around the nostrils was carefully removed with forceps for access to odorants, and the individual immersed in fresh system water. The dish was placed under a 20 X (NA = 0.5) water immersion objective on a Zeiss LSM 980 confocal microscope. Images were acquired in *XYT* acquisition mode with 512 × 200 pixel resolution at a rate of 2 Hz and digitized eight bit from single focal planes. The vehicle control (deionized water) followed by the odorant solution (0.2 ml) was slowly expelled through plastic tubing (Tygon R-3606; 0.8 mm ID, 2.4 mm OD) attached to a 1 ml syringe (BD 309659) and with the other end positioned in front of an individual's face. Two sequential applications of each were spaced by 1 min intervals. To calculate fluorescence intensity, regions of interest (ROI) were manually drawn around each cell in the average focal plane with the polygon tool and ROI *manage* in Fiji (*Schindelin et al., 2012*). To normalize calcium activity for each neuron to baseline fluorescence, the fractional change in fluorescence (ΔF/F) was calculated before the application of odorants (average of 320 frames from each neuron), according to the formula $F = (F_i - F_0)/F_0$, where $F_i$ is the fluorescence intensity at a single time point and $F_0$ is the baseline fluorescence. All data and images were analyzed using custom programs in MATLAB (MathWorks, version 7.3), Excel software and GraphPad Prism9.

## Assay of *Fos* expression in adult zebrafish

Individual adult zebrafish (7–9 months old) were placed in a tank with 1 L system water and acclimated for at least 1 hr prior to odorant exposure. Each odorant solution (1 ml) was gently pipetted into the tank water and the fish was kept there as the odorant diffused. After 30 min, the fish was sacrificed in an ice water slurry, and the brain dissected out and fixed in 4% paraformaldehyde in 1 X PBS overnight at 4 °C. Fixed brains were embedded in 4% low melting agarose in 1 X PBS and sectioned at 50 µm (for juvenile brains) or 70 µm (for adult brains) using a vibratome (VT1000S, Leica Biosystems, Inc). For more precise counting of *fos* expressing cells in adult brains, habenular sections were 35 µm thick. Sections were covered in 50% glycerol in 1 X PBS under coverslips. Bright-field images were captured with a Zeiss AxioCam HRc camera mounted on a Zeiss Axioskop. A Leica SP5 confocal microscope was used for fluorescent images. Data from *fos* RNA in situ hybridization experiments were quantified using ImageJ/Fiji software (*Schindelin et al., 2012*).

## Behavioral assays

Behavioral assays were performed using 5–7 week-old juvenile zebrafish and adults that were of 4–8 months of age. Responses to odorants were measured between 10:00 a.m. and 4:00 p.m. and fish were starved for 1 day prior to testing (*Koide et al., 2009*). Individual adults were placed in a 1.5 L test tank (Aquatic Habitats) in 1 L of system water and allowed to acclimate for at least 1 hr. For experiments with juveniles, individuals were acclimated to the behavior room for 1 hr, gently netted into the test tank (20 × 9 x 8.3 cm, 1.5 L mating cage) containing 0.6 L system water and maintained there for 5 min prior to testing. Swimming activity was recorded for 5 min (4 min for juveniles) before and after the application of odorants. Odorants (2 ml for adults, 1 ml for juveniles) were slowly expelled through plastic tubing (Tygon R-3606; 0.8 mm ID, 2.4 mm OD) attached on one end to a 3 ml syringe (BD 309657) and on the other positioned at one end of the test tank. A preference index was calculated using the formula: Preference to odorant = [(Total time spent in the half of the tank where odorant was delivered) − (Total time spent in the other half of the tank)]/Total time (*Koide et al., 2009*; *Wakisaka et al., 2017*). A preference index of –1 indicates that the position of an individual fish is maximally distanced from the application site of the odorant.

## Quantification and statistical analyses

Data were collected using custom written scripts in MATLAB (MathWorks). Statistical analyses were performed using GraphPad Prism9 (GraphPad Software, Inc). The Wilcoxon signed-rank test was used to compare a set of matched samples. An unpaired *t*-test was used to compare two independent data sets and two-way ANOVA followed by Bonferroni's post hoc test was used to compare datasets greater than two. All tests were two-tailed and results depicted as non-significant (*ns*, $p > 0.05$) or significant (*, $p < 0.05$, **$p < 0.01$, ***$p < 0.001$, and ****$p < 0.0001$).

## Acknowledgements

We are grateful to Bethany Malskis for animal care and to Dr. Krishan Ariyasiri and Dr. Eugene Demidenko for advice on statistical analyses. We thank Dr. Shin-ichi Higashijima for the *Gbait-hsp70:Gal4* donor plasmid, Dr. Wenbiao Chen for *pT3TS nCas9n* plasmid, Dr. Koichi Kawakami for the *UAS:zBoTXBLC-GFP* construct, Paul Krieg for the *Xla.Tubb* promoter construct, Dr. Claire Wyart for *GCaMP6f* plasmid and Dr. Wolfgang Driever and Dr. Tatjana Piotrowski for providing *bsx*[m1376] and *tcf7l2*[zf55] mutant zebrafish, respectively. This work was supported by NIH grants R37HD091280 and R01HD078220 to MEH.

## Additional information

### Funding

| Funder | Grant reference number | Author |
|---|---|---|
| National Institutes of Health | R01HD078220 | Marnie E Halpern |
| National Institutes of Health | R37HD091280 | Marnie E Halpern |

The funders had no role in study design, data collection and interpretation, or the decision to submit the work for publication.

### Author contributions

Jung-Hwa Choi, Conceptualization, Data curation, Formal analysis, Investigation, Methodology, Performed all of the experiments, Validation, Visualization, Writing – original draft, Writing – review and editing; Erik R Duboue, Methodology, Software, Writing – review and editing, Wrote MATLAB script for analyzing behavioral experiments; Michelle Macurak, Methodology, Resources, Technical assistance, constructed sgRNAs for lratd2a and slc5a7a; Jean-Michel Chanchu, Methodology, Resources, Technical assistance, generated Tol2 constructs and transgenic lines; Marnie E Halpern, Conceptualization, Funding acquisition, Methodology, Project administration, Resources, Supervision, Writing – review and editing

### Author ORCIDs

Jung-Hwa Choi http://orcid.org/0000-0002-0591-9708
Erik R Duboue http://orcid.org/0000-0003-3303-5149
Marnie E Halpern http://orcid.org/0000-0002-3634-9058

### Ethics

All zebrafish protocols were approved by the Institutional Animal Care and Use Committee (IACUC) of the Carnegie Institution for Science [Protocol #122] or Dartmouth College [Protocol #00002253(m3a)].

### Decision letter and Author response

Decision letter https://doi.org/10.7554/eLife.72345.sa1
Author response https://doi.org/10.7554/eLife.72345.sa2

## Additional files

### Supplementary files

• Supplementary file 1. Primers for making plasmids using the Gateway cloning system.

• Transparent reporting form

• Source code 1. Example code file to run the MATLAB script.

• Source code 2. MATLAB script used for behavioral analyses and for running the example code file (Source Code File 1).

### Data availability

All data generated or analyzed during this study are included in the manuscript and supporting file. Source data files have been provided for Figures 1-5. Behavioral analyses were performed using custom written scripts in MATLAB and uploaded as Source code Files.

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
