## [Editor Report]

This work presents a conceptual advance on our understanding of the habenula in vertebrate species, by revealing interesting functions to specific cell types within this region of the brain.

---

## [Decision Letter]

**Decision letter after peer review:**

Thank you for submitting your article "Specialized neurons in the right habenula mediate response to aversive olfactory cues" for consideration by *eLife*. Your article has been reviewed by 3 peer reviewers, one of whom is a member of our Board of Reviewing Editors, and the evaluation has been overseen by Marianne Bronner as the Senior Editor. The reviewers have discussed their reviews with one another, and the Reviewing Editor has drafted this to help you prepare a revised submission.

Essential revisions:

The three reviewers have appreciated the novelty and originality of the study, but note that improved visualization, quantifications and statistical analyses will be necessary to fully support the conclusions of the manuscript. Without performing these quantifications and statistical tests for all figures as detailed below, the magnitude and significance of reported effects are not clear, nor do they take into account the variability of the measures and the dependence of some of the measures.

1. Anatomy (Figure 1):

The authors should verify the specificity of the novel transgenic lines generated in the study to target lratd2a+ neurons in the right dHb (some left dHb expression is seen in Figure 1C-E). In addition, improving the quantification of c-fos and lratd2a overlap is necessary.

2. Calcium imaging and response to chemical stimuli (Figure 2):

The effects of the aversive chemical stimuli on the calcium activity of the lratd2a+ cells in the dorsal habenula are not clear in the traces presented nor quantified for all cells in all fish recorded. As a matter of fact, the traces of single examples show responses to water alone and not a very clear effect of the olfactory cues. The statistics are lacking to compare the response to water, cadaverine and chondroitin sulfate. The authors should determine whether the olfactory stimulus used induced a larger response than water in the same cells. They should assess the variability across fish, and the reliability of the effects. We advise to use linear mixed models which are suited to take into account the variability of cells within the same fish, as well as the effect of clutch or day of the experiment. In order to to take into account the effect of time and pairing within the same cell that receives the water stimuli before the olfactory ones, we recommend using a linear mixed model with a fixed effect of time (measuring the average or max DFF before, during and after the chemical stimulation) and treatment (olfactory cue added), pairing values within the same cell and same fish.

3. Behavior (Figures 3 and 4):

The effects of genetic manipulations on the response to cadaverine and chondroitin sulfate should be quantified and compared across genotypes using a proper statistical tests. In the submitted manuscript, the authors only quantified an effect of the drug *within* each group using t-tests using only the last point before the drug was applied, instead of comparing *between* the groups the effects of different genotypes on the preference index at all time points (before and after the application of the olfactory cue). This is an issue for both figures, note that in Figure 4 the preference index during baseline might differ between the wild type and mutant group. The authors should use a two-way repeated measures analysis of variance (ANOVA), or Kruskal-Wallis test for non-parametric data, to assess differences between treatment groups over time for validation studies, and test the effect of the drug over time *within* each group and *between* conditions before and after. By doing so, the authors will determine whether there is an effect of time, whether there is an effect of genotype and if there is an effect of time, measure the effect of an interaction. Note that if the authors opt for the ANOVA, they should perform post hoc comparison tests using either the Tukey method (commonly used to make pairwise comparisons) by correcting for multiple testing or t-tests with Bonferroni correction for multiple comparison. By doing so, the authors will determine whether there is an effect of time, whether there is an effect of genotype and measure the effect of a possible interaction.

*Reviewer #1 (Recommendations for the authors):*

Figure 1: Anatomy

To show that lratd2a+ neurons receive inputs from the olfactory bulb, the authors rely on gross low scale overlap of YFP+ axons on the soma of mApple-CAAX in the triple transgenic line lratd2a:QF2; QUAS:mApple-CAAX; lhx2a:YFP,

– A: First thing, add a quantification of (a) the number of neurons that are expressing lratd2a in the wild type fish at 7, 14, 21dpf and adult stage and (b) the number of neurons labeled in the lratd2a:QF2 transgenic line and the left and right side as it is central to the study – in the C panel, there seems to be expression of the transgene obtained by KI in the lratd2a locus (lratd2a:QF2)c601 in the left side.

– Next, add histological examinations or higher fluorescent images for panels D-D".

– Finally, can you specify if the overlap has been observed at the level of single planes or projections as it should be to suggest direct connectivity? If so, add a quantification to illustrate the overlap based on single planes from optical sections obtained on confocal with high mag and resolution and close up.

Figure 2: Calcium Imaging

– A: in order to understand whether the response of lratd2a+ neurons is specific to aversive cues: can you show the response to (a) water (as illustrated in B), (b) one other aversive cue as well as (c) one non aversive cues?

– A: for 14 dpf and response to chondroitin sulfate, please add the number of neurons missing.

– A: for 21 dpf and response to chondroitin sulfate, how come there are only 16 neurons measured instead of ~>35 ? Does it mean that the line is variegated and numbers can vary from 15 to 40 in the right habenula?

– B: since spontaneous activity is large and we cannot conclude for single examples, panels need to show the variability across cells and not only single trace : (a) use mean and ste to show the intrinsic variability of the response across cells and (b) quantify the response by calculating the peak and subtracting the baseline averaged over a similar time window before stimulation.

– C: the response to cadaverine measured with cfos is not clearly overlapping with expression of lratd2a: the authors should perform double fluorescent in situ labeling to demonstrate the overlap at the cellular level.

Figure 3: Behavior of adult zebrafish expressing botulinum toxin in a subset of neurons in the right dorsal habenula.

– A-E: the authors should validate that the intersectional genetics combining a KI line, the QF2/QUAS and Cre under the scl5a7a promoter is effective to target the lratd2a neurons only and which proportion of them. On these images, it appears that the KI line has expression on the left Habenula as well.

– G-J: The effect of the expression of the botulinum toxin is not clear at all (for the preference index: T-test a single time points on a subset of them can be misleading) statistics need to be improved : if the data is quantified and plotted every minute, we would expect to compare the conditions before baseline and establish that there is no difference in preference index before addition of cadaverine, and that a difference is observed after the addition and quantify for how long.

For the alarm substance, the data is not represented the same as for cadaverine: in the single measure of before/after, there is no difference across genotypes in speed, onset time of the fast swim, or time between fast swimming and freezing. But would there be a difference for cadaverine using the same single measure of before/after ? Probably not.

The statistics and choice of parameters needs to be sorted out and represented fully and fairly with consistency across compounds and figures.

Figure 4: Behavior of left-isomerized adult zebrafish.

– Expression of lratd2a is affected in the dorsal right and ventral right and left habenula so the mutant does not reveal only the role of lratd2a+ neurons in the dorsal right locus.

– Same issue here for the behavior: the pre-condition appears possibly different for the homozygous mutant and control sibling. The authors should test whether there are any difference of preference index in the two groups before drug application, and after drug application.

– Why are again different parameters plotted in H-G for the alarm substance compared to cadaverine? In addition instead of time onset and interval between fast and freeze, duration in the top of the tank is quantified. This choice looks arbitrary and all parameters should be chosen and kept the same for comparing the effects of cadaverine and alarm substance.

*Reviewer #2 (Recommendations for the authors):*

1. If calcium imaging experiments were to be repeated, water and odorant cues should be alternated so a direct comparison can be made for individual neurons. Also, both sides of the habenula could be simultaneously imaged, with lratd2a neurons labeled with another (e.g. RFP) marker, to allow for comparison between left and right habenular responses.

2. Can c-fos experiments be performed on the lratd2a transgenic background in adults to facilitate quantification?

3. While not necessary for this paper, chemogenetic approaches (e.g. TRPV1 from Prober lab) could be useful to activate the population

*Reviewer #3 (Recommendations for the authors):*

The authors of this work describe how cholinergic neurons expressing the lratd2a gene of the right dHb increase their activity to aversive odorant guiding aversive behaviors. The design of the study is very elegant, especially exciting is the combination of genetic tools that allow to label, as well as manipulate synaptic function. The authors present elegantly their data, and my impression is that this work deserves publication. The author may want to consider the following points:

1. The concepts of aversion and avoidance are confusing. Avoidance implies a form of learning (see reviews from J LeDoux) after an individual learns to anticipate an upcoming aversive stimulus. If I correctly interpreted the authors use a on-line reading of escape/aversive behaviour after inclusion of cadaverine. This should be probably better defined throughout the text.

2. At one point in the results the authors make use of a genetic approach allowing to control synaptic function with Botulinun neurotoxin. They state that "Axons labeled by BoTxBLC-GFP terminated at the vIPN.…. suggesting that botulinum neurotoxin inhibits synaptic transmission within this restricted region of the vIPN". I understand the technology is used in published data, yet it would be elegant to show along with the behavioural results an assessment of collapsed synaptic function.

3. The authors use a labeling strategy that allows claiming that the cholinergic neurons innervate a precise area of the IPN, supporting previous data in literature. In their data set the authors however study the functional responses to cadaverine only in somata of these neurons. It would be extremely relevat in my opinion to show that calcium transients are also detectable in the Cholinergic axons in the IPN. This would corroborate the functional integration of aversive signal within this neuronal circuit, and not only within the right habenula.

4. The authors may want to consider referring to the following reviews when citing habenula work in rats, mice, and humans in the context of physiology and disease: Hailan Hu et al., 2020; Lecca et al., 2014; Proulx et al., 2014.

[Editors' note: further revisions were suggested prior to acceptance, as described below.]

Thank you for resubmitting your work entitled "Specialized neurons in the right habenula mediate response to aversive olfactory cues" for further consideration by *eLife*. Your revised article has been reviewed by 2 peer reviewers and the evaluation has been overseen by Marianne Bronner as the Senior Editor, and a Reviewing Editor.

The authors show an interesting role of a subpopulation of habenular neurons in the avoidance response to aversive olfactory cues. We thank the authors for improving the manuscript, particularly for repeating the calcium imaging experiments and for improving the data presentation. There are however some remaining issues that need to be addressed, as outlined below.

Essential revisions:

1) Overall the behavioral differences to cadaverine shown in Figure 3G (Cre- vs Cre+) are relatively mild, especially as the aversion indices are significantly different from baseline in both Cre- and Cre+ condition (Figure 3—figure supplement 2B). Given that this is a key experiment in the paper, a discussion regarding whether this is a limitation of the existing tools (e.g. insufficient neurons silenced) or a reflection of underlying biology (e.g. redundancy in circuits for avoidance, different circuits controlling duration vs magnitude of aversion) would be beneficial.

2) Presentation of Figure 2c-d can be improved further – the same neurons presumably are being imaged "before" and "after", however the way the data is currently plotted makes it look like they are independent neurons.

3) It is counterintuitive that a negative aversion index means stronger aversion (perhaps call it a preference index instead, or flip the signs so more positive = more aversive).

4) In the aversion assay (3G, 4E, 5F): can the authors clarify if some form of multiple comparisons correction was done in calculating the p-values at each time bin?

5) The authors have performed ANOVA on the aversion indices shown in the supplementary figures, and report a significant effect of odor and odor x group interaction. Is there a significant effect of group alone? there is no explicit mention of the aversion index in the main text, and no interpretation in the figure legends. For clarity, the authors should elaborate how the statistical results from this 2nd analysis method ties in with / complements the statistical methods used in the main figures.

*Reviewer #2 (Recommendations for the authors):*

I thank the authors for improving on the manuscript, particularly for repeating the calcium imaging experiments and for improving the data presentation.

While significant effort has been put into improving the statistics, I have some additional questions about the analyses performed and their interpretation. Ultimately I will defer to the other reviewers regarding whether they are satisfied with the current methods.

For the main figures, I was expecting a two-way ANOVA to be performed for the time course data in the aversion assay (3G, 4E, 5F) to compare main effects of group and time and group x time interactions. I understand the authors are using a different methodology here (signed rank test) which has also been applied in other papers – however, can I clarify if some form of multiple comparisons correction was done in calculating the p-values at each time bin?

The authors have performed ANOVA on the aversion indices shown in the supplementary figures, and report a significant effect of odor and odor x group interaction. I might have missed something, but is there a significant effect of group alone? I also do not see any explicit mention of the aversion index in the main text, and no interpretation in the figure legends. For clarity, perhaps the authors could elaborate how the statistical results from this 2nd analysis method ties in with / complements the statistical methods used in the main figures?

Overall the behavioral differences to cadaverine shown in Figure 3G (Cre- vs Cre+) are relatively mild, and the aversion indices are significantly different from baseline in both Cre- and Cre+ condition (Figure 3—figure supplement 2B). The data is what it is, but given that this is a key experiment in the paper, a discussion regarding whether this is a limitation of the existing tools (e.g. insufficient neurons silenced) or a reflection of underlying biology (e.g. redundancy in circuits for avoidance, different circuits controlling duration vs magnitude of aversion) could be refreshing.

*Reviewer #3 (Recommendations for the authors):*

All my specific points were addressed by the authors. The complementary experiments and the modification provided in the text improved the paper and it is in my view suitable for publication.

---

## [Author Response]

Essential revisions:The three reviewers have appreciated the novelty and originality of the study, but note that improved visualization, quantifications and statistical analyses will be necessary to fully support the conclusions of the manuscript. Without performing these quantifications and statistical tests for all figures as detailed below, the magnitude and significance of reported effects are not clear, nor do they take into account the variability of the measures and the dependence of some of the measures.

We thank the editors and reviewers for their positive feedback and for the care they took to assess our data analysis, which we agree was problematic. We corrected all of the statistical tests that were used as suggested by the reviewers and consulted with an expert in biostatistics, Dr. Eugene Demidenko, a Professor in the Department of Biomedical Data Science at the Geisel School of Medicine at Dartmouth, to ensure we were performing the analyses correctly. We also repeated the calcium imaging experiments on 7 day old larvae so that the same neurons were individually monitored for both olfactory cues and vehicle controls.

1. Anatomy (Figure 1):The authors should verify the specificity of the novel transgenic lines generated in the study to target lratd2a+ neurons in the right dHb (some left dHb expression is seen in Figure 1C-E). In addition, improving the quantification of c-fos and lratd2a overlap is necessary.

The *lratd2a* gene is asymmetrically expressed in the left and right dHb, with transcripts in more cells and far more abundant on the right than on the left as assayed by RNA in situ hybridization. Consistent with this finding, in the transgenic lines we generated, expression of the QF2 transcription factor under the control of endogenous *lratd2a* cis-regulatory sequences drives QUAS regulated reporter labeling in more neurons in the right dHb and at higher levels. We have added a new Figure showing the quantification of *lratd2a*-expressing neurons in the left and right dHb in the QF/QUAS transgenic larvae (Figure 1—figure supplement 2).

We now show magnified images of cells co-expressing *fos* and *lratd2a* in Figure 2 to improve visualization of the overlap in colorimetric double in situ hybridization. We attempted to perform these experiments with fluorescently labeled probes, but we were unable to achieve the sensitivity needed to detect both of these relatively low abundance transcripts in doubly labeled cells.

2. Calcium imaging and response to chemical stimuli (Figure 2):The effects of the aversive chemical stimuli on the calcium activity of the lratd2a+ cells in the dorsal habenula are not clear in the traces presented nor quantified for all cells in all fish recorded. As a matter of fact, the traces of single examples show responses to water alone and not a very clear effect of the olfactory cues. The statistics are lacking to compare the response to water, cadaverine and chondroitin sulfate. The authors should determine whether the olfactory stimulus used induced a larger response than water in the same cells. They should assess the variability across fish, and the reliability of the effects. We advise to use linear mixed models which are suited to take into account the variability of cells within the same fish, as well as the effect of clutch or day of the experiment. In order to to take into account the effect of time and pairing within the same cell that receives the water stimuli before the olfactory ones, we recommend using a linear mixed model with a fixed effect of time (measuring the average or max DFF before, during and after the chemical stimulation) and treatment (olfactory cue added), pairing values within the same cell and same fish.

We agree with these suggestions and have performed new experiments in which we assess the change in GCaMP6f fluorescence in the same set of *lratd2a* right dHb neurons in response to the application of water or to odorant. We included the standard error of the mean to the traces to show the variability in the response among neurons (refer to Figure 2A-D).

3. Behavior (Figures 3 and 4):The effects of genetic manipulations on the response to cadaverine and chondroitin sulfate should be quantified and compared across genotypes using a proper statistical tests. In the submitted manuscript, the authors only quantified an effect of the drug within each group using t-tests using only the last point before the drug was applied, instead of comparing between the groups the effects of different genotypes on the preference index at all time points (before and after the application of the olfactory cue). This is an issue for both figures, note that in Figure 4 the preference index during baseline might differ between the wild type and mutant group. The authors should use a two-way repeated measures analysis of variance (ANOVA), or Kruskal-Wallis test for non-parametric data, to assess differences between treatment groups over time for validation studies, and test the effect of the drug over time within each group and between conditions before and after. By doing so, the authors will determine whether there is an effect of time, whether there is an effect of genotype and if there is an effect of time, measure the effect of an interaction. Note that if the authors opt for the ANOVA, they should perform post hoc comparison tests using either the Tukey method (commonly used to make pairwise comparisons) by correcting for multiple testing or t-tests with Bonferroni correction for multiple comparison. By doing so, the authors will determine whether there is an effect of time, whether there is an effect of genotype and measure the effect of a possible interaction.

We have shown responses to odorants within groups and between groups (refer to Figures 2-5, Figure 3—figure supplement 1-2, Figure 4—figure supplement 1 and Figure 5—figure supplement 1). We quantified and compared the response to odorants between groups during 5 min before and after application of cadaverine so the effects of genetic manipulations can be assessed (refer to Figure 3—figure supplement 2, Figure 4—figure supplement 1 and Figure 5—figure supplement 1).

As correctly pointed out, we have now performed the appropriate statistical analyses (refer to Figure legends). For analyzing response to cadaverine within a group, we used the Wilcoxon signed-rank test and paired t-test and cited publications that used these tests to evaluate the results of comparable behavioral experiments (Koide et al., 2009; Wakisaka et al., 2017). For analyzing between groups, we used two-way ANOVA followed by Bonferroni's post hoc test.

Reviewer #1 (Recommendations for the authors):Figure 1: AnatomyTo show that lratd2a+ neurons receive inputs from the olfactory bulb, the authors rely on gross low scale overlap of YFP+ axons on the soma of mApple-CAAX in the triple transgenic line lratd2a:QF2; QUAS:mApple-CAAX; lhx2a:YFP,– A: First thing, add a quantification of (a) the number of neurons that are expressing lratd2a in the wild type fish at 7, 14, 21dpf and adult stage and (b) the number of neurons labeled in the lratd2a:QF2 transgenic line and the left and right side as it is central to the study – in the C panel, there seems to be expression of the transgene obtained by KI in the lratd2a locus (lratd2a:QF2)c601 in the left side.

We have added a new Figure showing the quantification of *lratd2a*-expressing neurons in the left and right dHb in QF/QUAS transgenic larvae (Figure 1—figure supplement 2). Because of the morphology of the adult dHb, it is more challenging to obtain accurate cell numbers for the adult brain.

– Next, add histological examinations or higher fluorescent images for panels D-D".

As suggested, we have added higher magnification single plane confocal images from transverse sections of the adult brain (refer to new Figure 1—figure supplement 3B).

– Finally, can you specify if the overlap has been observed at the level of single planes or projections as it should be to suggest direct connectivity? If so, add a quantification to illustrate the overlap based on single planes from optical sections obtained on confocal with high mag and resolution and close up.

We provided single plane confocal images of the axon terminals of *lhx2a* neurons labeled with the presynaptic marker synaptophysin (refer to new Figure 1—figure supplement 3A). We also quantified the overlap in labeling of axon terminal and *lratd2a* neurons in confocal optical sections and now provide this information in the Figure 1—figure supplement 3.

Figure 2: Calcium Imaging– A: in order to understand whether the response of lratd2a+ neurons is specific to aversive cues: can you show the response to (a) water (as illustrated in B), (b) one other aversive cue as well as (c) one non aversive cues?

In this paper, we focused on the role of *lratd2a* right dHb neurons in mediating aversive cues as the dorsal habenulae has been implicated in fear-related behaviors (Agetsuma et al., 2010; Lee et al., 2010; Lee et al., 2019; Okamoto et al., 2012).

– A: for 14 dpf and response to chondroitin sulfate, please add the number of neurons missing.– A: for 21 dpf and response to chondroitin sulfate, how come there are only 16 neurons measured instead of ~>35 ? Does it mean that the line is variegated and numbers can vary from 15 to 40 in the right habenula?

The previous Figure 2 was removed because we repeated calcium imaging experiments (i.e., determined the change in GCaMP6f fluorescence in the same set of *lratd2a* right dHb neurons in response to the application of water or to odorant). The new results are shown in Figure 2A-D.

– B: since spontaneous activity is large and we cannot conclude for single examples, panels need to show the variability across cells and not only single trace : (a) use mean and ste to show the intrinsic variability of the response across cells and (b) quantify the response by calculating the peak and subtracting the baseline averaged over a similar time window before stimulation.

As suggested, each trace now includes the standard error of the mean to show the variability in the response among neurons. The new figures represent the fluorescent changes above baseline, which are averaged over a similar time window before and after application of water controls and of odorants (refer to Figure 2A and B). We also provide the complete data set for all neurons imaged over a 5 sec interval for two applications of vehicle alone followed by two applications of an odorant, each administered 1 min apart (new Figures 2C and D).

– C: the response to cadaverine measured with cfos is not clearly overlapping with expression of lratd2a: the authors should perform double fluorescent in situ labeling to demonstrate the overlap at the cellular level.

Magnified images of cells co-expressing *fos* and *lratd2a* are now shown in Figure 2 to allow better visualization of the overlap in labeling. Double fluorescent in situ hybridization was not sensitive enough to detect these transcripts.

Figure 3: Behavior of adult zebrafish expressing botulinum toxin in a subset of neurons in the right dorsal habenula.– A-E: the authors should validate that the intersectional genetics combining a KI line, the QF2/QUAS and Cre under the scl5a7a promoter is effective to target the lratd2a neurons only and which proportion of them. On these images, it appears that the KI line has expression on the left Habenula as well.

The *lratd2a* gene is asymmetrically expressed in the left and right dHb, but expression on the left is in fewer cells and at barely detectable levels. We quantified the number of dHb neurons labeled in the intersectional transgenic approach and included these data in the new (Figure 3—figure supplement 4B).

– G-J: The effect of the expression of the botulinum toxin is not clear at all (for the preference index: T-test a single time points on a subset of them can be misleading) statistics need to be improved : if the data is quantified and plotted every minute, we would expect to compare the conditions before baseline and establish that there is no difference in preference index before addition of cadaverine, and that a difference is observed after the addition and quantify for how long.The statistics and choice of parameters needs to be sorted out and represented fully and fairly with consistency across compounds and figures.

As pointed out by the reviewer, we corrected the statistical analyses for all behavioral data (please refer to the revised Figure legends). For analyzing the response to cadaverine within groups, we used the Wilcoxon signed-rank test and paired t-test and cited papers (Koide et al., 2009; Wakisaka et al., 2017) and for analyzing the responses between groups, we used two-way ANOVA followed by Bonferroni's post hoc test.

For the alarm substance, the data is not represented the same as for cadaverine: in the single measure of before/after, there is no difference across genotypes in speed, onset time of the fast swim, or time between fast swimming and freezing. But would there be a difference for cadaverine using the same single measure of before/after ? Probably not.

Please refer to the responses below for clarification on the changes that we have now made on depicting data from cadaverine and alarm substance. We have added Figures to be more consistent.

Figure 4: Behavior of left-isomerized adult zebrafish.– Expression of lratd2a is affected in the dorsal right and ventral right and left habenula so the mutant does not reveal only the role of lratd2a+ neurons in the dorsal right locus.

We had referred to this issue in the discussion: “Our findings also rule out a role for the ventral habenulae in the response to alarm substance, as the reaction to alarm substance was intact in transgenic adults in which *lratd2a* neurons were inactivated by *BoTxBLC* in the bilateral vHb as well as in the right dHb.” (page 15, lines 295-297)

– Same issue here for the behavior: the pre-condition appears possibly different for the homozygous mutant and control sibling. The authors should test whether there are any difference of preference index in the two groups before drug application, and after drug application.

We analyzed behavior during the entire experiment (i.e., 5 mins before and after addition of odorant). We have added a new figure that provides the data for all individual fish that were tested (refer to Figure 3—figure supplement 2, Figure 4—figure supplement 1 and Figure 5—figure supplement 1).

– Why are again different parameters plotted in H-G for the alarm substance compared to cadaverine? In addition instead of time onset and interval between fast and freeze, duration in the top of the tank is quantified. This choice looks arbitrary and all parameters should be chosen and kept the same for comparing the effects of cadaverine and alarm substance.

We agree that we could have been more consistent in presenting these results. We now show the same parameters for the response to alarm substance in BoTx-GFP and intersectional BoTx-GFP transgenic lines, and in *tcf7l2* and *bsx* mutants (refer to Figures 3, 4, 5 and Figure 3—figure supplement 1).

Reviewer #2 (Recommendations for the authors):1. If calcium imaging experiments were to be repeated, water and odorant cues should be alternated so a direct comparison can be made for individual neurons. Also, both sides of the habenula could be simultaneously imaged, with lratd2a neurons labeled with another (e.g. RFP) marker, to allow for comparison between left and right habenular responses.

We agree with the reviewers that it is important to compare responses to an odorants relative to the vehicle control in the same individual cells. We therefore include the results of the requested experiment performed on 7 dpf larvae (refer to new Figure 2A-D).

2. Can c-fos experiments be performed on the lratd2a transgenic background in adults to facilitate quantification?

Fluorescence in situ hybridization with the *fos* probe was not as sensitive as colorimetric in situ hybridization and we were not able to detect transcripts using this approach. We did not use a GFP probe to label *lratd2a* neurons in the transgenic background as we would still need to rely on a colorimetric reaction to detect *fos* transcripts.

3. While not necessary for this paper, chemogenetic approaches (e.g. TRPV1 from Prober lab) could be useful to activate the population.

We do not yet have this reagent under QUAS control, but it is a good idea to build this line for future work. We appreciate the suggestion.

Reviewer #3 (Recommendations for the authors):The authors of this work describe how cholinergic neurons expressing the lratd2a gene of the right dHb increase their activity to aversive odorant guiding aversive behaviors. The design of the study is very elegant, especially exciting is the combination of genetic tools that allow to label, as well as manipulate synaptic function. The authors present elegantly their data, and my impression is that this work deserves publication. The author may want to consider the following points:1. The concepts of aversion and avoidance are confusing. Avoidance implies a form of learning (see reviews from J LeDoux) after an individual learns to anticipate an upcoming aversive stimulus. If I correctly interpreted the authors use an on-line reading of escape/aversive behaviour after inclusion of cadaverine. This should be probably better defined throughout the text.

Thank you for this valuable feedback. We completely agree that avoidance could be a misleading term based on learning paradigms and have changed our language throughout to reflect aversion or repulsion, which gives a more accurate description of the observed behavioral response.

2. At one point in the results the authors make use of a genetic approach allowing to control synaptic function with Botulinun neurotoxin. They state that "Axons labeled by BoTxBLC-GFP terminated at the vIPN.…. suggesting that botulinum neurotoxin inhibits synaptic transmission within this restricted region of the vIPN". I understand the technology is used in published data, yet it would be elegant to show along with the behavioural results an assessment of collapsed synaptic function.

We didn’t examine neuronal activity at the axon terminals in the BoTxBLC-GFP transgenic line, however, we confirmed that expression of BoTxBLC-GFP inhibits the touch response (refer to Video 1). To measure transmission in dHb axon terminals at the vIPN effectively will require electrophysiology or more sensitive transgenic tools. We concur that this will be useful for future studies.

3. The authors use a labeling strategy that allows claiming that the cholinergic neurons innervate a precise area of the IPN, supporting previous data in literature. In their data set the authors however study the functional responses to cadaverine only in somata of these neurons. It would be extremely relevat in my opinion to show that calcium transients are also detectable in the Cholinergic axons in the IPN. This would corroborate the functional integration of aversive signal within this neuronal circuit, and not only within the right habenula.

A recent study (Zaupa et al., 2021) reported that “axon terminals of cholinergic and non-cholinergic habenular neurons exhibit stereotypic patterns of spontaneous activity that are negatively correlated and localize to discrete subregions of the target IPN” which has prompted us to consider the responses to odorants at the specific region of the IPN where the axons of *lratd2a* expressing neurons terminate. However, these axons are a small subset of the total cholinergic population and their terminals are challenging to visualize in vivo at larval stages. Such experiments will require us to adopt the explant approach used in Zaupa et al. or to develop transgenic lines to measure the responses of IPN neurons. As with the earlier point, we envision such strategies for our future experiments.

4. The authors may want to consider referring to the following reviews when citing habenula work in rats, mice, and humans in the context of physiology and disease: Hailan Hu et al., 2020; Lecca et al., 2014; Proulx et al., 2014.

The cited references describe functions of lateral/ventral habenulae. Our study focuses on the dorsal habenulae of zebrafish, which are equivalent to the medial not the lateral habenulae of rodents. We ruled out the involvement of vHb in this study as indicated in the discussion, thus it is not relevant to cite the published work on this other brain region.

[Editors' note: further revisions were suggested prior to acceptance, as described below.]

Essential revisions:1) Overall the behavioral differences to cadaverine shown in Figure 3G (Cre- vs Cre+) are relatively mild, especially as the aversion indices are significantly different from baseline in both Cre- and Cre+ condition (Figure 3—figure supplement 2B). Given that this is a key experiment in the paper, a discussion regarding whether this is a limitation of the existing tools (e.g. insufficient neurons silenced) or a reflection of underlying biology (e.g. redundancy in circuits for avoidance, different circuits controlling duration vs magnitude of aversion) would be beneficial.

As suggested, we have now elaborated on these points in the Discussion (page 14, line 281) as follows:

“As a group, zebrafish in which the synaptic activity of *lratd2a* neurons in the right dHb was inhibited by *BoTxBLC-GFP* did not exhibit repulsion to cadaverine, although some individuals showed a mild aversive response that was not sustained relative to sibling controls. […] More finely tuned techniques for temporal or spatial regulation of neuronal inactivation would help resolve these issues.”

2) Presentation of Figure 2c-d can be improved further – the same neurons presumably are being imaged "before" and "after", however the way the data is currently plotted makes it look like they are independent neurons.

We now present the data in revised figures that clearly indicate the results from the same individual neurons before and after addition of odorants or vehicle control (refer to Figures 2C and 2D).

3) It is counterintuitive that a negative aversion index means stronger aversion (perhaps call it a preference index instead, or flip the signs so more positive = more aversive).

We agree with this feedback. We changed the aversive index to a preference index and we explain how this was measured in the Methods and in the Results (refer to revised Figure 3—figure supplement 1 and Figures 3G, 4E, Figure 4—figure supplement 1, Figure 5F and Figure 5—figure supplement 1).

4) In the aversion assay (3G, 4E, 5F): can the authors clarify if some form of multiple comparisons correction was done in calculating the p-values at each time bin?

We now show new figures (3G, 4F and 5F) with the data analyzed by ANOVA followed by Bonferroni's correction. The previous Figures 3G, 4E and 5F were moved to Figure 3—figure supplement 1A, Figure 4—figure supplement 1A and Figure 5—figure supplement 1A. We believe that it is useful to include the statistical analyses of individuals within groups in supplemental figures as they reveal minute by minute behavioral differences not apparent in the ANOVA multiple comparisons.

5) The authors have performed ANOVA on the aversion indices shown in the supplementary figures, and report a significant effect of odor and odor x group interaction. Is there a significant effect of group alone?

As correctly pointed out by the reviewer, we had mistakenly omitted the effect of group alone although we had done these analyses. The effect of group values are now all provided in the figure legends (refer to legends for Figure 3—figure supplement 1B, Figure 4—figure supplement 1B and Figure 5—figure supplement 1B).

There is no explicit mention of the aversion index in the main text, and no interpretation in the figure legends.

We had previously included the description of the aversion index in the Material and Methods section of the main text. As suggested by Reviewer 2, we have modified this and now depict the results as a “preference index”. We have updated this information accordingly in the “Behavioral assays” section of the Materials and methods and we have also added the following sentence to the Results (page 7, line 141): “We used a preference index that is based on the position of an individual fish at a given time relative to the application site of the odorant (refer to Materials and methods).”

For clarity, the authors should elaborate how the statistical results from this 2nd analysis method ties in with / complements the statistical methods used in the main figures.

Depicting our results with different statistical approaches reveals different aspects of the aversive behavior we observed within and between groups of adults. We now have analyzed all data by ANOVA followed by Bonferroni's correction and provide the results in the main figures (3G, 4F and 5F). We moved the previous graphs to supplemental figures (Figure 3—figure supplement 1A, Figure 4—figure supplement 1A and Figure 5—figure supplement 1A). For all experiments, we show the raw data for the response to cadaverine over time and used the Wilcoxon signed-rank test to compare a set of matched samples (i.e., behavior before and after application of cadaverine by the same fish) within each group. We provide this information for all experimental paradigms in Figure 3—figure supplement 1A, Figure 4—figure supplement 1A and Figure 5—figure supplement 1A.

As recommended by the reviewer, we updated the following paragraphs in the Results section to describe how the results from different statistical methods tie in (page 9, line 174):

“To determine whether the *lratd2a* neurons in the right dHb contributed to the aversive response to cadaverine, we monitored the behavior within and between groups of adults that had or did not have the *BoTxBLC-GFP* transgene. Fish lacking the transgene showed a significantly reduced preference for the side of the tank where cadaverine had been applied (Figure 3G).

When the response of individual fish within each group was compared over time (Figure 3—figure supplement 1), adults with or without *BoTxBLC-GFP* initially avoided the side of the test tank where cadaverine had been introduced. However, aversion was sustained for 4 min in control fish, but not in those expressing *BoTxBLC-GFP* in *lratd2a* neurons. These findings, from statistical tests on individuals both within and between groups, suggest that *lratd2a* neurons in the right dHb are required for a prolonged aversive response to cadaverine.”